# Resilience Assessment of Interdependent Infrastructure Systems: A Case Study Based on Different Response Strategies

**Jingjing Kong [1], Slobodan P. Simonovic [1] and Chao Zhang [2],***

[1] Department of Civil and Environmental Engineering, The University of Western Ontario, London, ON N6A 5B9, Canada; kjj121123@gmail.com (J.K.); simonovic@uwo.ca (S.P.S.)
[2] Shanghai Key Laboratory of Financial Information Technology, Shanghai University of Finance and Economics, Shanghai 200433, China
* Correspondence: zhang.chao@sufe.edu.cn

**Abstract:** Resilient infrastructure systems are essential for continuous and reliable functioning of social and economic systems. Taking advantage of network theory, this paper models street network, water supply network, power grid and information infrastructure network as layers that are integrated into a multilayer network. The infrastructure interdependencies are described using five basic dependence patterns of fundamental network elements. Definitions of dynamic cascading failures and recovery mechanisms of infrastructure systems are also established. The main contribution of the paper is a new infrastructure network resilience measure capable of addressing complex infrastructure system, as well as network component (layer) interdependences. The new measure is based on infrastructure network performance, proactive absorptive capacity and reactive restorative capacity, with three resilience features of network—robustness, resourcefulness, and rapidity. The quantitative resilience measure using dynamic space-time simulation model is illustrated with a multilayer infrastructure network numerical test, including different response strategies to floods of different scale. The results demonstrate that the resilience measure provides an evaluation method of various protection and restoration strategies that will optimize the performance of interdependent infrastructure system. The sector-specific decisions could not always lead to optimal system solutions, and systems approach offers significant benefits for increasing infrastructure system resilience. This study can assist municipal decision makers in (i) better understanding the effects of different response strategies on the resilience of interdependent infrastructure system, and (ii) deciding which strategy should be adopted under different types of disasters.

**Keywords:** resilience; infrastructure system; multilayer network; infrastructure interdependence; restoration strategy

## 1. Introduction

Infrastructure systems, such as telecommunications, electric power systems, natural gas and oil, transport, and water supply, are essential for continuous and reliable functioning of social and economic systems, as they provide them with the fundamental services that support the economic productivity, security, and population quality of life. There are practical links between disaster risk management, global change adaptation and sustainable development leading to the reduction of disaster risk and re-enforcing resilience as a new development paradigm. There has been a noticeable change in the approaches to the management of disasters in many countries, moving from disaster vulnerability to disaster resilience; the latter viewed as a more proactive and positive expression

of community engagement with disaster risk management. With the increasing damage of various disasters to infrastructure systems, it is important to develop effective response strategies to maintain the sustainable development of the social and economic systems. Infrastructure systems are not isolated, but highly interconnected and mutually interdependent [1]. For example, water and telecommunication systems need a steady supply of electrical energy to maintain their normal operations, while electrical power systems require water (among other resources) and various telecommunication services for power generation and delivery. In reality, interdependencies can improve infrastructure operational efficiency. However, recent worldwide disasters have shown that interdependencies can increase system vulnerability as damage to one infrastructure system can produce cascading failures of others and result in disasters on the regional or national scales [2]. For example, Hurricane Sandy in 2012 caused enormous power utility damage which led to the non-operation of major fuel pipelines, telecommunication infrastructure and water and sewage facilities. Then the transportation system collapsed, due to electric and fuel outages, and the health care centers shut down as all the infrastructure systems did not function anymore, and New York was paralyzed [3]. Since the protection and recovery from system disturbances are quite often complicated and difficult, the resilience of the infrastructure systems is almost always overestimated and becomes a focal point for policy development [4–6]. There is a clear need for further study of critical interconnected infrastructure resilience.

The concept of resilience has recently received significant attention of both, researchers and practitioners in various disciplines. Its origins are in ecology and work of C. S. Holling [7]. Due to the wide interest and application in various disciplines, there is neither universal definition nor widely accepted general quantitative approach for its assessment. Excellent reviews are available in a number of published papers [8–11]. Infrastructure resilience is always seen as the ability of the designed distributed and interconnected systems: (i) To reduce the magnitude of impact and/or duration of disruptive events [12]; (ii) resist (prevent and withstand) any possible hazard; (iii) absorb the initial damage; and (iv) recover to normal function [6]. According to the published literature on infrastructure resilience, the metrics are always outcome-oriented [9]. Most researchers use a synonymous metric of system robustness or adverse effect of system vulnerability [13–15] as a measure of resilience. Their focus is on extent or duration of component failures that a system can tolerate for an individual infrastructure type [16,17]. Some studies quantify resilience from absorptive, adaptive, and restorative capacity perspectives [18,19].

Infrastructure systems taking the form of networks abound in the world, such as the Internet, transportation networks, water transport networks, etc. [20]. Network theory is one of the fundamental pillars of discrete mathematics, and has been used to explore the robustness, vulnerability and resilience of different infrastructure networks. Taking advantage of network theory structure analysis, infrastructure systems can be described as networks, where nodes represent infrastructure components (such as water pumps and electric transformers), and links mimic the physical and relational connections among different infrastructure components (such as electric tie lines and water pipes) [21–23]. The critical problem is that interdependencies among infrastructures result cascading failure, and make resilience evaluation more complicated.

Infrastructure system components may be dependent and interdependent in various ways—most interpretations from the earlier literature view interdependencies as macro-properties of coupled systems. The interdependencies can be characterized as either physical, cyber, geographic and logical. Based on these qualitative studies, infrastructure system interdependencies have been described with different models and analyzed using model simulations. Reed et al. [24] use a causal network diagram for illustrating the relationship between power delivery and the other infrastructures to describe the real performance of an infrastructure system. Ouyang et al. [25–27] introduced power-to-gas unidirectional interdependency to model the cascading failures. Filippini et al. [28] illustrate the functional relationships among interdepend infrastructure components to present the power and gas infrastructures' structural and dynamic properties. There are other approaches to address interdependencies among multi infrastructures, but most of them are focused on the cascading

failures within and across multiple systems to estimate system-level vulnerability, with only a few putting emphases on recovery processes to evaluate system-level resilience [26,29]. Few published studies analyze the whole process resilience of integral infrastructure system with interdependences, and seldom address resilience from proactive absorptive capacity and reactive restorative capacity perspective [29].

In this paper, a generic network model resilience is developed based on the definition from [15], which is derived from the work of MCEER (Multidisciplinary Center for Earthquake Engineering Research) [30,31]. To apply this time-dependent resilience definition to interdependent infrastructure network systems requires: (i) Development of network system performance curves during a specific time period from the beginning of the disturbance until the time of full system recovery; and (ii) description of joint performance of multiple infrastructure network layers with the consideration of cascading failures. The resilience model of integrated infrastructure network is developed to capture both, proactive absorptive capacity and reactive restorative capacity, with the consideration of detailed interdependences among diverse infrastructure components. The approach might be used for quantitative infrastructure system resilience evaluation, development and selection of restoration strategies, and development and testing of preparedness, response and recovery plans.

The remainder of the paper is organized as follows. The second part of the paper presents a multilayer infrastructure network model, and formulates basic dependence patterns of individual infrastructure components that are capable of addressing dynamic cascading failures and various recovery strategies. Next, the multilayer infrastructure network resilience is defined, and quantitative resilience metric is presented that provides for understanding the dynamics of multilayer infrastructure networks under disturbances. Afterwards, five infrastructure system restoration strategies focusing on different interdependence patterns are introduced. Next, the new resilience model use is illustrated with a multilayer infrastructure network numerical test, including different response strategies to floods of different scale. The paper ends with the discussion of the multiple impacts of diverse interdependences on infrastructure network and conclusions.

## 2. Infrastructure Network Formalization

Infrastructure systems play an important role in assuring the health and vitality of the social and economic fabric of the city or country. Individual infrastructure systems, such as the system of streets or power grid, function together as a "system of systems," in which two or more infrastructure types interact with one another [32,33]. A system-of-systems can be described by a topology that accounts for the representation of its components and the way they interact. Infrastructure systems (defined here as the system of public works of a country, state, or region), with diverse clearly defined components, can be modeled as a multilayer network [21,34].

### 2.1. Infrastructure Network Representation

The infrastructure network model is based on the network theory, where two basic components, nodes and edges, build up the model of a system. A network is always represented by $G$, the nodes set and edges set are represented by $N$ and $E$, respectively. This paper focuses on the main urban infrastructure system networks, including streets, power grid, water supply network, and information infrastructure.

In the case of the street network, each crossing and end point is represented by a node, whereas edges represent street segments. The street-layer network can be represented as $G^S(N^S, E^S)$, where $N^S$ is the set of street junctions and end points, and $E^S$ is the set of undirected street segments [35]. The superscript $s$ is the initial of the infrastructure type ($s$ for street network). The edges are undirected and homogeneous. The nodes can be regarded as parts of the adjacent street segments. Generally, the street network is fully connected, there is a path linking any pair of nodes, and there are multiple paths. A path will disappear if any of its edges are destroyed.

Water supply network (system of intakes, reservoirs, pumping stations, pipelines, conduits, etc., by which water is collected, purified, stored, and pumped to an urban area) is represented as $G^W(N^W, E^W)$, where intakes, reservoirs and pumping stations are denoted as nodes with different attributes and water distribution pipes and conduits are denoted as edges [36,37]. Different from the street network, edges of water supply network are directed as the water flow from an intake to pump stations and storage facilities through distribution pipes. Generally, the water supply networks are represented as trees without a circle and redundant edges. The downstream nodes and edges could not operate unless all the upstream nodes and edges function normally.

Power grid network is represented as $G^P(N^P, E^P)$, where power plants and transformer substations are denoted as nodes with different attributes, and power lines are denoted as directed edges [38]. Same as the water supply network, edges of power grid are directed as the electricity is transmitted from power plants to transformer substations, and then distributed through power lines. The downstream nodes and edges could not operate unless all the upstream nodes and edges function normally [39,40]. In the case of power grid networks, different from the water supply networks, there are redundant power lines connecting important distribution and transformer substations.

The information infrastructure network is represented as $G^I(N^I, E^I)$, where Internet service providers are denoted using nodes, and cable connections are denoted as undirected edges [41]. As the exchange of information is bidirectional, the edges are undirected. According to scale, population and structure of a city, information network could be in different shapes—a star, chain or circle. A node or edge normally operate if the path connecting them with the source node exists.

All the individual type of infrastructure networks introduced above can be illustrated as single network layers. The infrastructure system model is a network of networks integrating all of them, and it may be presented as a multilayer network [21] as illustrated in Figure 1. Nodes and edges in the same layer belong to the same type of infrastructure (shown as the same color solid lines within a single layer network in Figure 1). Edges between different layers denote interdependences between different types of infrastructure (shown as dotted lines between different layers in Figure 1). The color of an edge identifies the direction of dependency. For example, the red dotted line between nodes belonging to the power grid and water supply network illustrates the electricity from the electric infrastructure provided to water supply infrastructure. The blue dotted line between two networks illustrates water transfer from the water infrastructure layer to power grid infrastructure (for example, for cooling the thermal power plants).

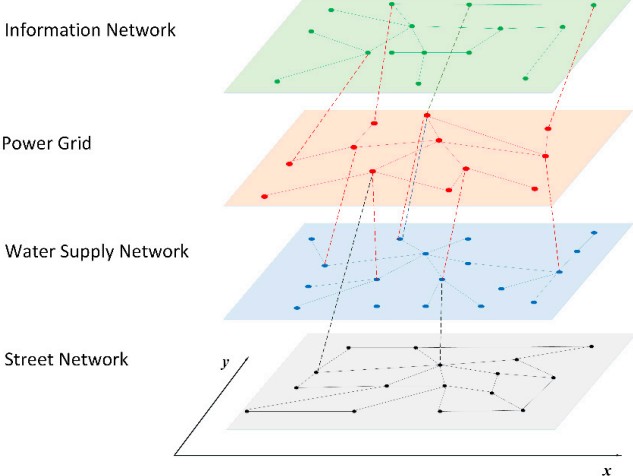

**Figure 1.** Interdependent infrastructure system model.

As different infrastructure components located in the same area may be subject to a specific disturbance/disaster, it is necessary to consider the location of infrastructure in the model description. Furthermore, the location of infrastructure has an important effect on topological properties and

consequently on infrastructure functioning processes [42]. Therefore, the spatial attributes of nodes and edges should be included in a realistic infrastructure network model with geographical coordinates, which can be defined in a two-dimensional Euclidean coordinate system. Therefore, each node has three coordinates $(\phi, x, y)$, where $\phi$ denotes the type of the infrastructure, and $(x, y)$ denote the geographical location of the node. Edges are denoted by the two adjacent nodes and can be divided into two types: (i) Intra-infrastructure connection, the same value of $\phi$; and (ii) inter-infrastructure connection, different $\phi$ value.

### 2.2. Basic Infrastructure Dependence Patterns

Interdependence indicates the bidirectional interaction, which includes two directed dependencies between two infrastructure elements [1]. Urban infrastructure components can be dependent and interdependent in various ways. Most of the earlier literature view interdependencies as macro-properties of coupled systems classified in different ways [43]. Generally, not any components malfunction of one infrastructure system can result in efficiency reduction, function loss or system destruction of another macro-interdependent infrastructure system. Therefore, micro-structure, or basic pattern of infrastructure dependence, need to be considered. The focus of the proposed resilience model is the direct impact of infrastructure malfunction, which is always seen as the first-order effect.

Let us consider a two-layer infrastructure networks $G^{\phi_1}$ and $G^{\phi_2}$, where $\phi_1 \neq \phi_2$. Every element of these two networks has two exclusive states: On (normal operation) and off, (non-operation and malfunction). Normal operation refers to the designed level of operation. Non-operation indicates physical destruction of elements by a disturbance, and malfunction describes the situation when elements are not physically destroyed, but could not function, due to related resources outage. With different relations between four fundamental network structure elements: Nodes, edges, paths and clusters (combinations of nodes and edges), there are five basic infrastructure dependence patterns (four patterns are illustrated in Figure 2):

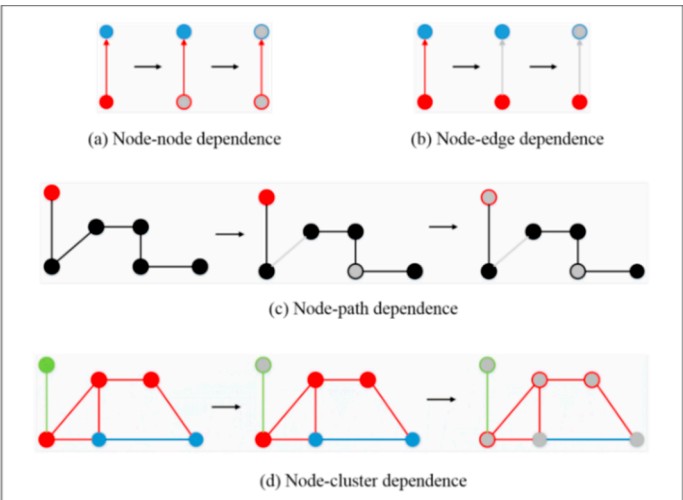

(a) Node-node dependence　　　　(b) Node-edge dependence

(c) Node-path dependence

(d) Node-cluster dependence

- Nodes and edges with blue, red, black and green represent water, electric, street and communication infrastructures respectively;
- Grey inside represent the malfunction state of the infrastructures;
- Arrows represent the time change.

**Figure 2.** Basic infrastructure dependence patterns.

(i) Node—node dependence (Figure 2a): The state of node $n_i^{\phi_1}$ is dependent on the state of $n_j^{\phi_2}$ via resource, service and information flows between them, or vice versa. For example, the state of water pump depends on the state of its connecting electric transmission substation. This pattern is represented as:

$$ID^{nn} = e_{ij}^{\phi_1 \phi_2} \tag{1}$$

where $ID^{nn}$ is node to node dependence between two networks, and $e_{ij}^{\phi_1\phi_2}$ is state of the edge linking the two nodes: Node *I* of $\phi_1$ network and node *j* of $\phi_2$ network.

(ii) Node—edge dependence (Figure 2b): The state of node $n_l^{\phi_1}$ is dependent on the edge $e_{ij}^{\phi_2}$ or $(n_i^{\phi_2}, n_j^{\phi_2})$, or vice versa. For example, the state of Internet service provider depends on its connecting power supply. This pattern is represented as:

$$ID^{ne} = n_l^{\phi_1} \times e_{ij}^{\phi_2} \tag{2}$$

where $ID^{ne}$ is node edge dependence, or vice versa. $n_l^{\phi_1}$ is the state of node *l* of $\phi_1$ network, and $e_{ij}^{\phi_2}$ is the state of edge of $\phi_2$ network linking nodes *I* and *j*.

(iii) Node/Edge—path dependence (Figure 2c): The state of node $n_l^{\phi_1}$ or edge $e_{lk}^{\phi_1}$ is dependent on the state of the path $p_{im}^{\phi_2}$, which is represented as $\{n_i^{\phi_2}, e_{ij}^{\phi_2}, n_j^{\phi_2}, e_{jk}^{\phi_2}, n_k^{\phi_2}, \ldots, n_m^{\phi_2}\}$. For example, the state of coal power plant is dependent on the path (transportation network) connecting the plant with coal supply locations. This pattern is represented as

$$ID^{NP} = n_l^{\phi_1} \times p_{im}^{\phi_2} \quad or \quad ID^{EP} = e_{lk}^{\phi_1} \times p_{im}^{\phi_2} \tag{3}$$

where $ID^{NP}$ is node path dependence $n_l^{\phi_1}$ is the state of node *l* of $\phi_1$ network, and $p_{im}^{\phi_2}$ is the state of path connecting node *i* and node *m* of $\phi_2$ network, and $p_{im}^{\phi_2} = \prod_i^m n_i^{\phi_2} \prod_{i,j}^{\cdot,m} e_{i,j}^{\phi_2}$. $ID^{EP}$ is edge another network path dependence, $e_{lk}^{\phi_1}$ is the state of edge linking nodes *l* and *k* of $\phi_1$ network.

(iv) Node/Edge—cluster dependence (Figure 2d): The state of cluster $c_i^{\phi_1}$, which is a set of nodes and their edges of network $G^{\phi_1}$, is dependent on the state of node $n_l^{\phi_2}$ or edge $e_{lk}^{\phi_2}$ of network $G^{\phi_2}$. For example, the operations of water or power infrastructure with the same geographic attributes being controlled by an internet service provider. This pattern is represented as:

$$ID^{NC} = n_l^{\phi_2} \times c_i^{\phi_1} \quad or \quad ID^{EC} = e_{lk}^{\phi_2} \times c_i^{\phi_1} \tag{4}$$

where $ID^{NC}$ is cluster node dependence. $n_l^{\phi_2}$ is the state of node *l* of $\phi_2$ network, and $c_i^{\phi_1}$ is the state of cluster of $\phi_1$ network. $ID^{EC}$ is cluster network edge dependence, $e_{lk}^{\phi_2}$ is the state of edge linking nodes *l* and *k* of $\phi_2$ network. $n_i^{\phi_1}$ and $e_{ij}^{\phi_1}$ are the state of elements of the cluster $c_i^{\phi_1}$, and $c_i^{\phi_1} = \prod n_i^{\phi_1} e_{ij}^{\phi_1}, (n_i^{\phi_1} \in c_i^{\phi_1}, e_{ij}^{\phi_1} \in c_i^{\phi_1})$.

(v) Geographic dependence: The state of all infrastructure elements located at the same location A are affected by a disturbance simultaneously. This pattern is represented as:

$$ID^{GL} = \left\{ n_i^{\phi_1} \cup e_{jk}^{\phi_1} \cup \cdots \cup n_p^{\phi_1} \cup e_{qr}^{\phi_1} \right\} \tag{5}$$

where $ID^{GL}$ is geographic dependence among nodes and edges. $n_i^{\phi_1}, e_{jk}^{\phi_1}, \ldots, n_p^{\phi_1}, e_{qr}^{\phi_1}$ are nodes and edges with coordinate values $(x, y)$ belonging to the same area A.

In the previous discussion we looked at two infrastructure networks. The basic dependence patterns can cause cascading impacts throughout the multilayer network as time goes on. Given three individual infrastructure networks $G^{\phi_1}, G^{\phi_2}, G^{\phi_3}$ ($\phi_1 \neq \phi_2 \neq \phi_3$), there are many combinations of the five basic dependence patterns, which could form chain or cycle reaction among three single layer networks and cause cascading failure spreading throughout the whole infrastructure system. On the other hand, interdependences could accelerate mitigation and be conducive to disturbance response with the repair of several components. They can contribute to strengthening system robustness and resilience with local protection.

### 2.3. Infrastructure System Dynamic Mechanism

The consequences of infrastructure system disturbances are usually twofold, the magnitude of interrupted services, and the duration of interruption [6]. In practice, the response of infrastructure to a disturbance can be expressed as: *Absorptive capacity*—the ability of infrastructure to absorb the impacts of disturbance and minimize the consequences with limited effort (i.e., buffering) [44]; or *restorative capacity*—the ability of infrastructure to recover from large-scale disaster with external or unconventional resources. These two response modes together define the *adaptive capacity* of the infrastructure [45]. The response of the infrastructure system to a disturbance varies with time—adding dynamic properties to interdependent infrastructure networks [29]. As an illustration of impacts and duration of disturbance, the adaptive capacity of $i$th infrastructure of $G^\phi AC_i^\phi$ can be represented as:

$$AC_i^\phi = IPD(performance_1, duration_1; performance_2, duration_2 \ldots) \tag{6}$$

where $performance_\bullet$ and $duration_\bullet$ represent the state of an infrastructure and the duration of modified performance, respectively. $AC_i^\phi$ is a function of these two properties.

As indicated in Section 2.2, each infrastructure can be in one of two states: Function and malfunction—represented with value of 1 or 0. Then the adaptive capacity of an infrastructure type, shown in Figure 3 can be represented as:

$$AC_i^\phi = (1, t_1 - t_0; 0, t_i - t_3; 1, t_{i+1}) \tag{7}$$

where $t_0$ is the time when the disturbance occurs, $t_1 - t_0$ is the buffering time $T_B$ calculated as the period of the infrastructure malfunctioning, $t_3$ is the starting time of infrastructure repair, $t_i - t_3$ is the repair time $T_R$, and $t_i - t_1$ is the malfunction time of the infrastructure $T_M$.

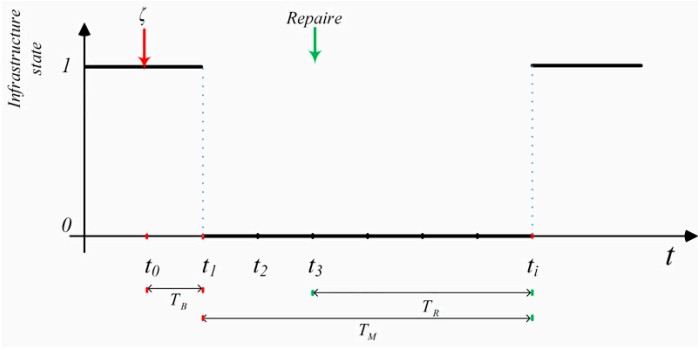

**Figure 3.** Performance of the infrastructure after a disturbance.

The adaptive capacity of a single layer network is a function of the adaptive capacity of all elements', the network size, structure and operation mechanism (intra-network interdependencies). Each adaptive layer capacity is obtained as the average value of the adaptive capacities of all elements obtained by dynamic integration. The adaptive capacity of multilayer infrastructure network is the function of the adaptive capacity of all single layer networks and inter-network interdependencies. Similarly, the adaptive capacity of multilayer network is obtained as the average value of adaptive capacities of all single layer networks obtained by dynamic integration. Therefore, the adaptive capacities of all single layer networks and the multilayer network are all numbers between 0 and 1.

As interdependences exist within elements of a single network layer and among multi-network layers, an infrastructure will malfunction if its dependent infrastructure is malfunctioning or if it is destroyed by the disaster event. Based on the basic infrastructure dependence patterns introduces in Section 2.2, the dynamic process of an infrastructure system (change of infrastructure state with time) affected by a disturbance can be described using a flow diagram in Figure 4.

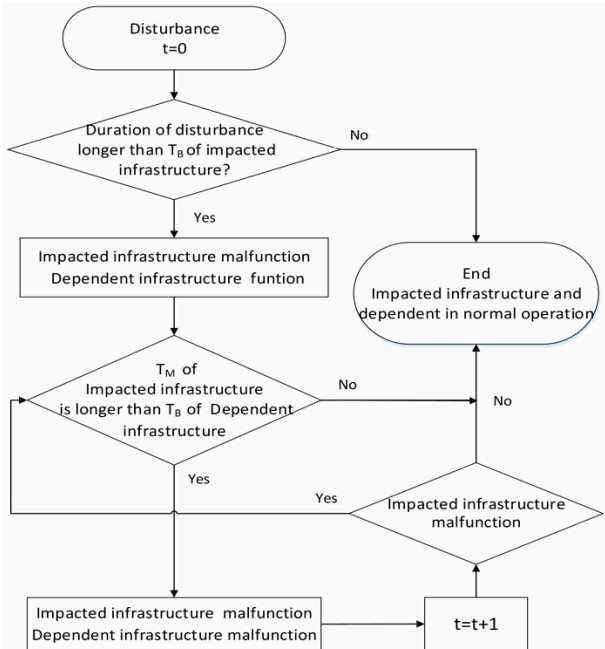

**Figure 4.** Dynamic process of an interdependent Infrastructure system after a disturbance.

Infrastructure in Figure 4 can be a node, an edge, a path or a cluster of an individual network layer (one type of infrastructure). For the node-node, node-edge and node/edge-cluster dependence modes, infrastructures are influenced by the state of their dependent components. For the node/edge-path dependence mode, an infrastructure is modeled as a node and an edge influenced by combined impacts of all the elements of the path, i.e., mixed states of the nodes and edges simultaneously. The infrastructure will return to normal operation when its dependent infrastructures are recovered. That is why (a) the quick response is essential for preventing massive loss of the infrastructure system and minimizing the impact on the society; and (b) why different response and restoration strategies would result in different impacts.

## 3. Infrastructure System Resilience Model

This section presents a new generic resilience model for the analyses of infrastructure performance under various conditions.

### 3.1. Infrastructure System Resilience Definition

The infrastructure system resilience is defined as the ability to prepare for, and adapt to changing conditions, and withstand and recover rapidly from disruptions, including the ability to withstand and recover from deliberate attacks, accidents, or naturally occurring threats or incidents [46,47]. Therefore, infrastructure system resilience includes two basic concepts: System performance and its adaptive capacity. The system resilience can be divided into two different forms: Proactive absorptive capacity and reactive restorative capacity. Proactive absorptive capacity mainly depends on the infrastructure (or network) inherent features, such as robustness, which can be determined in the infrastructure planning and design, or disaster preparation and mitigation period. Reactive restorative capacity mainly depends on the restorative strategies implemented during the operation period after the disturbance. This kind of classification is more practical for resilience improvement. Since the number of functioning infrastructure elements and the number of available resources are the foundations for recovery, the former capacity will directly influence the latter. In Equation (6), the adaptive capacity is defined using dynamic system performance, which is similar to the quality function of the resilience model derived by MCEER (Multidisciplinary Center for Earthquake Engineering Research) [30]. System performance curve includes the capacity to withstand and recover (see Figure 5).

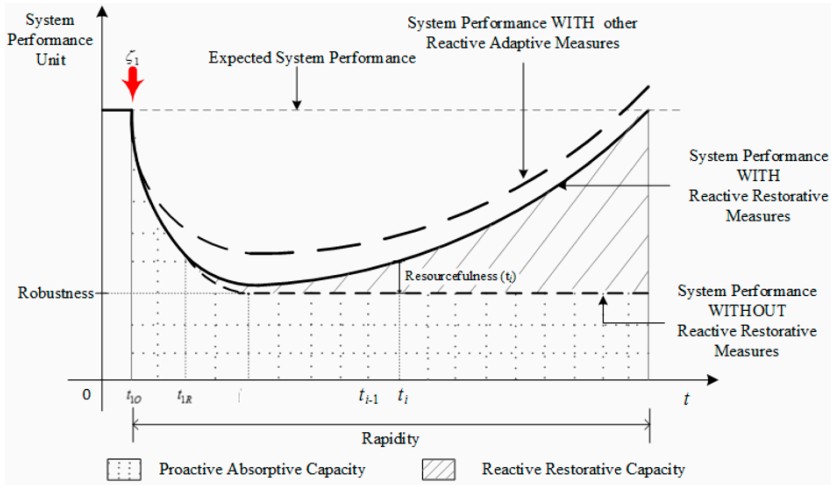

**Figure 5.** The typical performance of an infrastructure system with a disturbance.

System performance unit in Figure 5 is the infrastructure impact specific. It could be [km] of unblocked street length, [km$^3$] of water distribution volume, [kw] of power transmission capacity and [GB] of Internet traffic information volume. According to Figure 5, infrastructure systems may have the same performance and the same resilience, but one could be due to high robustness and low rapidity, while the other could be due to low robustness and high rapidity [48]. The former is reactive, absorptive capacity; the latter is proactive restorative capacity.

Infrastructure system, a typical "systems of systems", is a set of multiple and independently operational systems (such as street network, water supply network and power supply network, etc.) interacting with one another to meet specific needs [49]. Therefore, infrastructure system resilience refers not only to the ability to resist disturbance and reorganize intra-layer network layers, while undergoing change, but also the ability to retain essentially the same function, structure and feedbacks among inter-layer networks. The former capacity is important for an individual infrastructure system. The latter requires more systematic thinking and management, since it may be affected by small, unforeseen disturbances [50].

### 3.2. Three Features of Infrastructure System Resilience

System resilience is represented and quantified by four key features: Robustness, redundancy, resourcefulness, and rapidity [30]. Generally, redundancy can be considered at the component and network levels. Component redundancy includes buffer and slack resources, which can improve infrastructure reliability and adaptive capacity. Network redundancy includes structure duplication and structure modularity. Both redundancies are captured by the shape of the system performance curve before external restoration, and provide a measure of disturbance absorptive capacity and resistant capacity. Robustness is always used to evaluate the maximum level of disturbance resistance or tolerant capacity of an infrastructure system. Here redundancy is seen as one element of robustness. Therefore, there are three remaining features necessary for the implementation with infrastructure network systems.

#### 3.2.1. Robustness

Robustness refers to the ability of systems to withstand a given level of stress without suffering degradation or loss of function. The common measure for network robustness is the critical fraction of disturbance at which the system completely collapses [10]. For individual infrastructure, the robustness

is computed as the ratio of the minimum number of operational elements after a disturbance $\zeta_1$ to the total number of elements of one infrastructure type (one network layer), and repressed as $R_{Rob}^{\phi,\zeta_1}(t_{1R}^{\phi})$:

$$R_{Rob}^{\phi,\zeta_1}(t_{1R}^{\phi}) = \frac{n_o^{\phi}(t_{1R}^{\phi}) + e_o^{\phi}(t_{1R}^{\phi})}{N^{\phi} + E^{\phi}}, \tag{8}$$

where $t_{1R}^{\phi}$ is the time when the sum of $n_o^{\phi}(t)$ and $e_o^{\phi}(t)$ generates the maximum loss of a system $\phi$ performance after the disturbance $\zeta_1$, $n_o^{\phi}$ is the number of operational nodes and $e_o^{\phi}$ is the number of operational edges, $N^{\phi}$ and $E^{\phi}$ are the total numbers of nodes and edges, respectively.

For multilayer infrastructure network, the robustness is computed as the ratio of the minimum number of operational elements to the total number of elements over all network types (layers), which is shown in Figure 5, and repressed as $R_{Rob}^{\zeta_1}(t_{1R})$:

$$R_{Rob}^{\zeta_1}(t_{1R}) = \frac{\sum\limits_{\phi} (n_o^{\phi}(t_{1R}) + e_o^{\phi}(t_{1R}))}{\sum\limits_{\phi} (N^{\phi} + E^{\phi})}, \tag{9}$$

where $t_{1R}$ is the time when the sum of $n_o^{\phi}(t_{1R})$ and $e_o^{\phi}(t_{1R})$ of all single layer networks generates the maximum loss of performance after the disturbance $\zeta_1$, $n_o^{\phi}(t_{1R})$ is the number of operational nodes and $e_o^{\phi}(t_{1R})$ is the number of operational edges of a single layer network $\phi$ at $t_{1R}$, $N^{\phi}$ and $E^{\phi}$ are the total numbers of nodes and edges of single layer network $\phi$, respectively.

### 3.2.2. Resourcefulness

Resourcefulness is the capacity to develop and implement mitigation and response strategies to a specific disturbance. It is limited by the ability to obtain sufficient monetary, physical, technological, informational and human resources necessary to meet established priorities. In this work, the network performance of restoration strategies to a specific disturbance $\zeta_1$ is used for quantifying resourcefulness, repressed as $R_{Res}^{\phi,\zeta_1}(t)$:

$$R_{Res}^{\phi,\zeta_1}(t) = SP^{\phi,\zeta_1}(t) - SP_0^{\phi,\zeta_1}(t) = f\left(SP^{\phi,\zeta_1}(t-1), RS^{\phi,\zeta_1}(t-1)\right) - SP_0^{\phi,\zeta_1}(t), \tag{10}$$

where $SP^{\phi,\zeta_1}(t)$ is the system performance of network $\phi$ with restoration starting at $t$, $SP_0^{\phi,\zeta_1}(t)$ is the system performance of network $\phi$ without restoration at $t$, $RS^{\phi,\zeta_1}(t-1)$ is the response strategy of infrastructure network $\phi$ after disturbance $\zeta_1$ implemented at $t-1$. $f(\bullet)$ is the result of $SP^{\phi,\zeta_1}(t-1)$ and $RS^{\phi,\zeta_1}(t-1)$.

For multilayer infrastructure network, resourcefulness is shown in Figure 5 and quantified as $R_{Res}^{\zeta_1}(t)$:

$$R_{Res}^{\zeta_1}(t) = SP^{\zeta_1}(t) - SP_0^{\zeta_1}(t) = F\left(\sum SP^{\phi,\zeta_1}(t-1), \sum RS^{\phi,\zeta_1}(t-1)\right) - SP_0^{\zeta_1}(t), \tag{11}$$

where $SP^{\zeta_1}(t)$ is the system performance of multilayer network with restoration starting at $t$, $SP_0^{\zeta_1}(t)$ is the system performance of multilayer network without restorations at $t$, $\sum SP^{\phi,\zeta_1}(t-1)$ is the integration of all single layer networks performance at $t-1$, $\sum RS^{\phi,\zeta_1}(t-1)$ is combined restorative strategies of all utility sectors at $t-1$.

### 3.2.3. Rapidity

Rapidity refers to the capacity to meet priorities and achieve goals in a timely manner in order to minimize losses and avoid future infrastructure system disruptions. Duration of system recovery to normal operational levels is always used as a measure to evaluate system resilience, and can be seen as

the main figure-of-merit to evaluate proactive restorative capacity. In this research, the duration of system recovery is used to denote rapidity, repressed as $R_{Rap}^{\phi,\zeta_1}$:

$$R_{Rap}^{\phi,\zeta_1} = t_{1RE}^{\phi} - t_{1O'}^{\phi} \tag{12}$$

or

$$R_{Rap}^{\phi,\zeta_1} = t_{2O}^{\phi} - t_{1O'}^{\phi} \tag{13}$$

where $t_{1RE}^{\phi}$ is the time when infrastructure network $\phi$ recovers to the operational level equal to the one before the disturbance event $\zeta_1$, $t_{1O}^{\phi}$ is the occurrence time of the disturbance $\zeta_1$. $t_{2O}^{\phi}$ is the occurrence time of the disturbance $\zeta_2$. Equation (12) is used in situations with one disturbance event and Equation (13) in the case of sequential disturbances.

For multilayer infrastructure network, rapidity is computed as the longest duration of disturbance impacts of all the individual infrastructure network layers, due to the specific disturbance $\zeta_1$, which is shown in Figure 5, and represented as $R_{Rap}^{\zeta_1}$:

$$R_{Rap}^{\zeta_1} = \max\left\{R_{Rap}^{\phi,\zeta_1}\right\}. \tag{14}$$

### 3.3. Dynamic Infrastructure System Resilience Metric

Infrastructure network system performance and its adaptive capacity can fully describe the dynamic system behavior in response to system disturbance and implementation of various adaptation/restoration measures. Original Space-Time Dynamic Resilience Measure developed by Simonovic and Peck (2013) is adopted in this research to complex network infrastructure systems. It quantifies resilience as the difference between the area under expected system performance and actual system performance. Applying this measure, typical resilience of an infrastructure system with a disturbance is shown in Figure 6. The introduction of system adaptation measures provides for the increase in system resilience (by increasing the line shaded area in Figure 5). The system performance without adaptation measures is shown by the dashed line and with adaptation measures as a full line in Figure 5. The adaptive capacity can be achieved by proactive absorptive measures and reactive restorative measures.

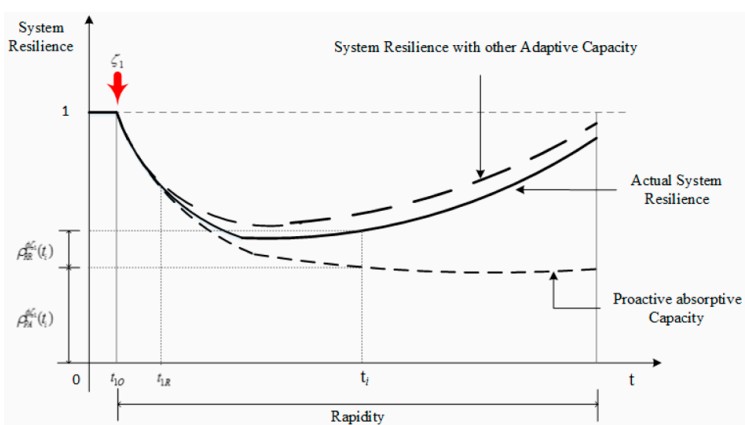

**Figure 6.** Typical resilience of infrastructure system with a disturbance.

Using the definition of the three features of resilience, the proactive absorptive capacity of individual infrastructure network subject to a disturbance $\zeta_1$ (represented as the dotted shaded area in Figure 5) can be calculated as $\rho_{PA}^{\phi,\zeta_1}(T)$:

$$\rho_{PA}^{\phi,\zeta_1}(T) = \frac{\int_{t_{1O}}^{T} SP_0^{\phi,\zeta_1}(t)dt}{1 \times T} = \frac{\int_{t_{1O}}^{t_{1R}^{\phi}} SP_0^{\phi,\zeta_1}(t)dt + \int_{t_{1O}}^{t_{1R}^{\phi}} R_{Rob}^{\phi,\zeta_1}(t_{1R}^{\phi})dt}{1 \times T}, t_{1O} \leq T \leq t_{1RE}^{\phi}, \tag{15}$$

where $SP_0^{\phi,\zeta_1}(t)$ is the system performance of the network $\phi$ after disturbance $\zeta_1$ without restoration strategy, which is calculated as the ratio of functional infrastructure network elements to the total number of elements of the network $\phi$ at $t$; and 1 in the denominator refers to the undisturbed system performance. After $t_{1R}^{\phi}$, $SP_0^{\phi,\zeta_1}(t)$ would be stable and equals to $R_{Rob}^{\phi,\zeta_1}(t_{1R}^{\phi})$, which is the robustness of network $\phi$ after disturbance $\zeta_1$. Then *Proactive absorptive capacity* of individual infrastructure network could be improved with its robustness increase.

For a multilayer infrastructure network, the proactive absorptive capacity metric can be described as $\rho_{PA}^{\zeta_1}(T)$:

$$\rho_{PA}^{\zeta_1}(T) = \frac{\int_{t_{1O}}^{T} \int_{\phi} SP_0^{\phi,\zeta_1}(t)dt}{1 \times T} = \frac{\int_{t_{1O}}^{t_{1R}} \int_{\phi} SP_0^{\zeta_1}(t)dt + \int_{t_{1O}}^{t_{1R}} \int_{\phi} R_{Rob}^{\zeta_1}(t_{1R})dt}{1 \times T}, t_{1O} \leq T \leq t_{1RE}^{\phi}. \tag{16}$$

The proactive absorptive capacity of multilayer infrastructure network could also be improved with its robustness increase, and be reduced with its robustness decrease.

The reactive restorative capacity of individual infrastructure network subject to a disturbance $\zeta_1$ (illustrated as the line shaded area in Figure 5) can be calculated as $\rho_{RR}^{\phi,\zeta_1}(T)$:

$$\rho_{RR}^{\phi,\zeta_1}(T) = \frac{\int_{t_{1O}}^{T} R_{Res}^{\phi,\zeta_1}(t)dt}{1 \times T}, \ t_{1O} \leq T \leq t_{1RE}^{\phi}. \tag{17}$$

For a multilayer infrastructure system network, the reactive restorative capacity metric $\rho_{PR}^{\zeta_1}(T)$ is:

$$\rho_{RR}^{\zeta_1}(T) = \frac{\int_{t_{1O}}^{T} \int_{\phi} R_{Res}^{\zeta_1}(t)dt}{1 \times T}, t_{1O} \leq T \leq t_{1RE}^{\phi}. \tag{18}$$

Reactive restorative capacity of individual infrastructure network or multilayer infrastructure system network would be improved with the increase of network resourcefulness or shorten of rapidity.

The resilience metric of a single layer infrastructure network $\phi$ under infrastructure network disturbance $\zeta_1$ can be now determined as $r^{\phi,\zeta_1}(T)$:

$$\begin{aligned} r^{\phi,\zeta_1}(T) &= \rho_{PA}^{\phi,\zeta_1}(T) + \rho_{RR}^{\phi,\zeta_1}(T) = \frac{\int_{t_{1O}}^{T} SP_0^{\phi,\zeta_1}(t)dt}{1 \times T} + \frac{\int_{t_{1O}}^{T} R_{Res}^{\phi,\zeta_1}(t)dt}{1 \times T} \\ &= \frac{\int_{t_{1O}}^{t_{1R}} SP_0^{\phi,\zeta_1}(t)dt + \int_{t_{1O}}^{t_{1R}} R_{Rob}^{\phi,\zeta_1}(t_{1R})dt + \int_{t_{1O}}^{T} R_{Res}^{\phi,\zeta_1}(t)dt}{1 \times T} \end{aligned} , \ t_{1O} \leq T \leq t_{1RE}^{\phi}. \tag{19}$$

The resilience of the multilayer infrastructure network under infrastructure network disturbance $\zeta_1$ can be now determined as $r^{\zeta_1}(T)$:

$$\begin{aligned} r^{\zeta_1}(T) &= \rho_{PA}^{\zeta_1}(T) + \rho_{RR}^{\zeta_1}(T) = \frac{\int_{\phi} \int_{t_{1O}}^{T} SP_0^{\phi,\zeta_1}(t)dt}{1 \times T} + \frac{\int_{\phi} \int_{t_{1O}}^{T} R_{Res}^{\zeta_1}(t)dt}{1 \times T} \\ &= \frac{\int_{t_{1O}}^{t_{1R}} \int_{\phi} SP_0^{\phi,\zeta_1}(t)dt + \int_{t_{1O}}^{t_{1R}} \int_{\phi} R_{Rob}^{\phi,\zeta_1}(t_{1R})dt + \int_{t_{1O}}^{T} \int_{\phi} R_{Res}^{\phi,\zeta_1}(t)dt}{1 \times T} \end{aligned} , \ t_{1O} \leq T \leq t_{1RE}^{\phi}. \tag{20}$$

The resilience of single or multilayer infrastructure network under disturbance $\zeta_1$ would be improved by the increase of proactive absorptive capacity or reactive restorative capacity. Further, resilience would be improved by the increase of robustness or network resourcefulness and shorten of rapidity.

The resilience metrics in Equations (19) and (20) are derived for a single infrastructure network disturbance. However, the exposure of urban infrastructure networks to various potential types of

disturbances may lead to the situation that the system experiences multiple disturbances, one after another, without sufficient time between them that is necessary for a full recovery. Under a sequential set of system disturbances $\zeta_1, \zeta_2, \ldots \zeta_d$, a single layer and a multilayer infrastructure system resilience $r^{\phi;\zeta_1,\zeta_2\ldots\zeta_d}(T)$ and $r^{\zeta_1,\zeta_2\ldots\zeta_d}(T)$ can be calculated as:

$$r^{\phi;\zeta_1,\zeta_2\ldots\zeta_d}(T) = \rho_{PA}^{\phi;\zeta_1,\zeta_2\ldots\zeta_d}(T) + \rho_{RR}^{\phi;\zeta_1,\zeta_2\ldots\zeta_d}(T) = \frac{\int_{t_{1O}}^{T} \int_{\zeta} SP_0^{\zeta}(t)dt}{1 \times T} + \frac{\int_{t_{1O}}^{T} \int_{\zeta} R_{Res}^{\zeta}(t)dt}{1 \times T},$$
$$t_{1O} \leq T \leq t_{1RE}^{\phi} \tag{21}$$

$$r^{\zeta_1,\zeta_2\ldots\zeta_d}(T) = \rho_{PA}^{\zeta_1,\zeta_2\ldots\zeta_d}(T) + \rho_{RR}^{\zeta_1,\zeta_2\ldots\zeta_d}(T) = \frac{\int_{t_{1O}}^{T} \int_{\zeta} \int_{\phi} SP_0^{\phi,\zeta}(t)dt}{1 \times T} + \frac{\int_{t_{1O}}^{T} \int_{\zeta} \int_{\phi} R_{Res}^{\phi,\zeta}(t)dt}{1 \times T},$$
$$t_{1O} \leq T \leq t_{1RE}^{\phi} \tag{22}$$

I"s worth noting that system robustness under subsequent disturbances deteriorates with time and each new disturbance event even if the direct physical impacts are the same. This is because the time between sequential disturbances is almost always shorter than the time needed for full system recovery.

According to Equation (22), the infrastructure system resilience can be improved through the increase of system proactive absorptive capacity $\rho_{PA}^{\zeta_1,\zeta_2\ldots\zeta_d}$ or reactive restorative capacity $\rho_{RR}^{\zeta_1,\zeta_2\ldots\zeta_d}$. Based on the definition of proactive absorptive capacity described by Equation (15), it is clear that this value is an inherent system characteristic and cannot be changed in a short period of time, especially after the disturbance event. *Proactive absorptive capacity* depends on pre-disaster planning and preparedness, which can be evaluated by robustness. Robustness is a function of network redundancy and component reliability. *Reactive restorative capacity* is flexible and depends on the adaptation/restoration strategies selected for implementation. Therefore, the reactive restorative capacity can be improved through timely response, effective response and systematic recovery. The following section of the paper concentrates on the relationship between the infrastructure system resilience and the restoration strategies which are taking into consideration the infrastructure system interdependences.

## 4. Restoration Strategies of Infrastructure System

Restoration strategies influence system resilience through the change of resourcefulness and rapidity. Under these practical limitations, the allocation of restoration resources, especially the repair sequence, gains in importance. In reality, after a disaster event, the last damaged infrastructure is always repaired first because of the accessibility to earlier damaged one (for example, damaged street segments located in the outside areas are repaired earlier than the damaged ones located in the epicenter area). On the other side, the first recovery of earlier damaged infrastructures can reduce the duration of impact on the early affected population. For each individual infrastructure system, there is the best restoration strategy to achieve a specific goal like, for example, restoration of power supply in shortest possible time for all residents. Sometimes, particular electric transmission substations get the priorities, because they provide support for critical infrastructure (like airports, hospitals, etc.). Unfortunately, in this process, hidden interdependencies among different infrastructure types are often overlooked, which could impact the system performance and resilience significantly. There are a few restoration strategies implemented to improve multilayer infrastructure system after a disaster.

In the rest of this manuscript, the infrastructure repair sequence is used as the restoration strategy. Five restoration strategies (RSs) are modeled and discussed below:

(i) First repair first failures *(RS-FF)*—This strategy is usually used during the emergency when time and space may not be available for a more elaborate response. The *RS-FF* strategy does not take infrastructure dependence into account.

(ii) First repair last failures *(RS-FL)*—Same as the *RS-FF* strategy, this strategy is used in an emergency without taking under consideration the infrastructure interdependence.

(iii) First repair important components independently *(RS-IE)*—First repair important components independently also do not take infrastructure dependence into account. It is usually used to maximize the benefits of a single sector and is quite widely studied in the literature.

(iv) First repair the obvious dependent elements *(RS-OD)*—This strategy considers obvious or physical interdependences of different infrastructure types, including node-node, node-edge and node/edge-cluster dependence modes.

(v) First repair the hidden dependent elements *(RS-HD)*—The fifth restoration strategy *(RS-HD)* takes latent interdependence or cyber, geographic and logistical infrastructure interdependences into account. They can be usually illustrated as node/edge-path dependences.

Every restoration strategy depends on the mobilization of restoration resources, including crews, vehicles, and specialized equipment for infrastructure repair. There are three assumptions used in this paper regarding all restoration strategies: (i) One unit of resource refers to a repair team, including repair crews, vehicles, equipment and some replacement components; (ii) every unit of resource has the same effectiveness; and (iii) response and recovery of each damaged infrastructure requires one unit of resources. Generally, after any disaster, resources are quite often scarce when compared with the extent of destroyed infrastructure.

### 4.1. First Repair First Failures Strategy

In most cases, the response strategies have to be implemented with highest possible urgency, without complete information about the disaster. Therefore, the *RS-FF* is a commonly used strategy. The strategy allocates resources to the first destroyed infrastructure with the highest priority. Here, "first" is defined according to the time of direct infrastructure destruction. The priority sequence is determined by the scarcity of the infrastructure and the degree of destruction. Based on the number of available resources, different repair sequences can be implemented. The flow diagram for implementing the *RS-FF* is shown in Figure 7.

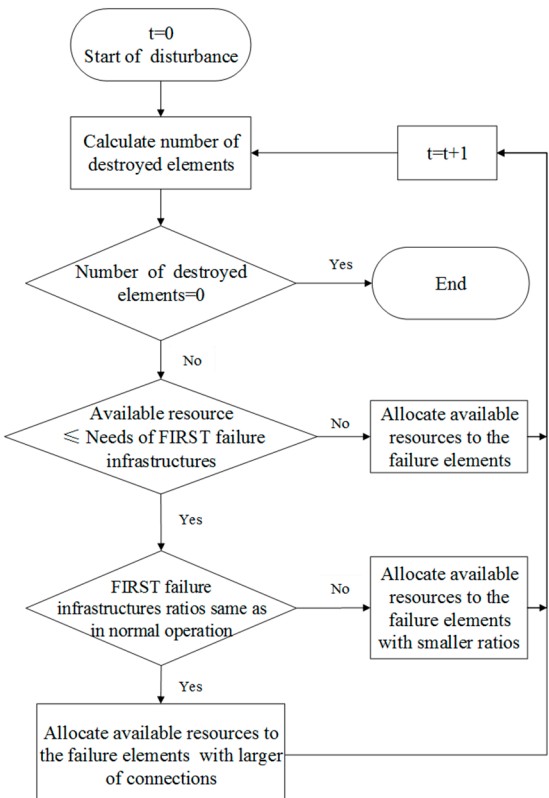

**Figure 7.** Flow diagram of RS-FF.

### 4.2. First Repair Last Failures Strategy

In cases when the areas affected by initial disturbance are totally destroyed or hard to reach, a first repair last failure strategy, *RS-FL*, may be implemented. According to *RS-FL*, the resources are allocated first to the last destroyed infrastructure of the highest priority. Here, "last" is defined by the time of direct infrastructure destruction by the disaster event, and priority is established according to the scarcity of the infrastructure and degree of destruction. Similar to the *RS-FF* strategy, different repair sequences can be implemented with limited restoration resources, and the decision process is shown in Figure 8.

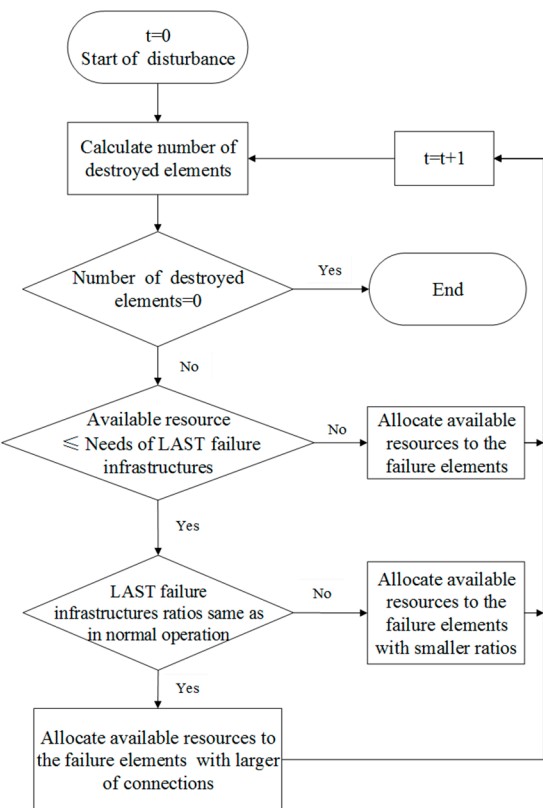

**Figure 8.** Flow diagram of RS-FL.

### 4.3. First Repair Important Components Independently

Operations, management and maintenance of individual infrastructure networks is the responsibility of different sectors. Decisions within each sector are driven by minimization of its own losses and rapid implementation of appropriate restoration strategies to achieve that. The *RS-IE* strategy starts first with the repair of damaged elements of the infrastructure of importance to individual infrastructure system. In the implementation of this step, the evaluation of the importance of each infrastructure element is a significant problem, and the number of elements required for return to normal operations is used as the decision measure. The decision process of *RS-IE* strategy is shown in Figure 9, which can be seen as the optimization of single layer network operations. The number of infrastructure types that can be repaired and sequence of repair is limited by the available restoration resources at each period, similar to *RS-FF* and *RE-FL* strategies. The resources are allocated randomly to the infrastructure elements with the same importance to its network. The importance of an infrastructure type is changing with time as a consequence of infrastructure connectivity.

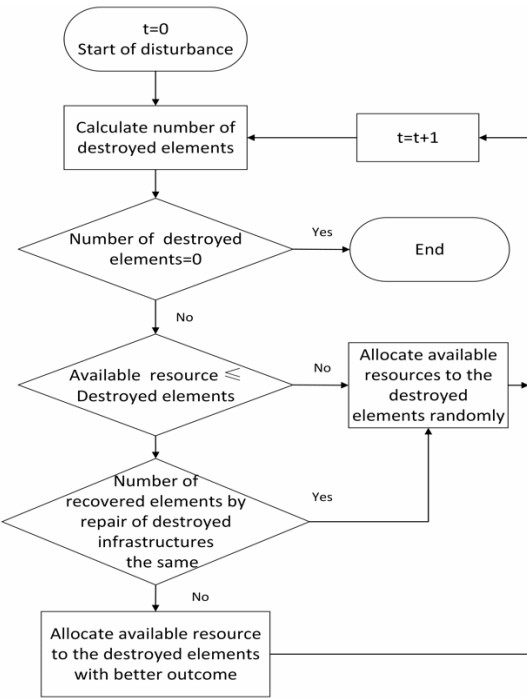

**Figure 9.** Flow diagram of RS- IE.

## 4.4. First Repair the Obvious Dependent Elements

More recent protection strategies are concerned with physical infrastructure interdependences and aimed at risk reduction. To improve infrastructure system restorative capacity, the restoration strategies also need to consider these infrastructure interdependences (illustrated with equations (1), (2) and (3)). As the interdependences between diverse infrastructure types (for example water pumps powered by electric power) are essential for normal functioning of dependent infrastructure, the initial step in the implementation of the *RS-OD* is to repair inter-network elements of higher importance during each time period. The importance of an inter-network infrastructure is measured by the proportional contribution of this kind of infrastructure to operational conditions without disturbance. The lower the proportion, the higher the importance. The *RS-OD* decision strategy is shown in Figure 10.

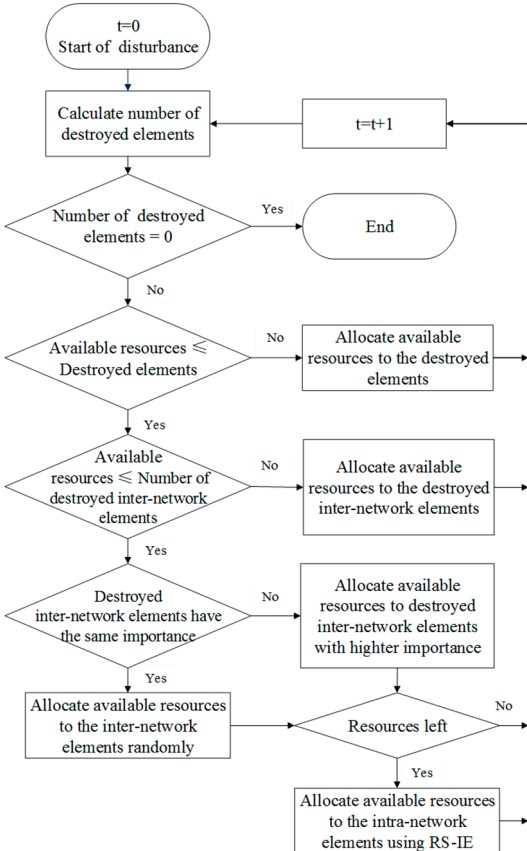

**Figure 10.** Flow diagram of RS-OD.

### 4.5. First Repair the Hidden Dependent Elements

Hidden dependencies, including functional and cyber dependence, represented by Equation (4), are not obvious in normal situations, and therefore, are easy to be ignored during disasters. Hidden dependencies can lead to huge losses as their malfunction can cause cascading failures of various infrastructure types. E.g., the destruction of streets connecting to the outside world may cause the malfunction of the power plant and potential collapse of the whole power grid, spreading to the malfunction of the water supply network and information network. Repair of the elements of these dependences is more beneficial for the whole infrastructure system recovery. Therefore, the fifth restoration strategy *RS-HD* considers hidden dependencies. The *RS-HD* starts first with the repair of hidden dependent elements (including nodes and edges of intra- and inter-network connections) of higher importance. The importance of hidden elements of dependent infrastructure is measured by the importance of the path, including these elements. The importance of the path is measured by the proportional representation of each element (i.e., nodes and edges) in normal operational conditions. The lower the proportion, the higher the importance. The *RS-HD* strategy is illustrated in Figure 11.

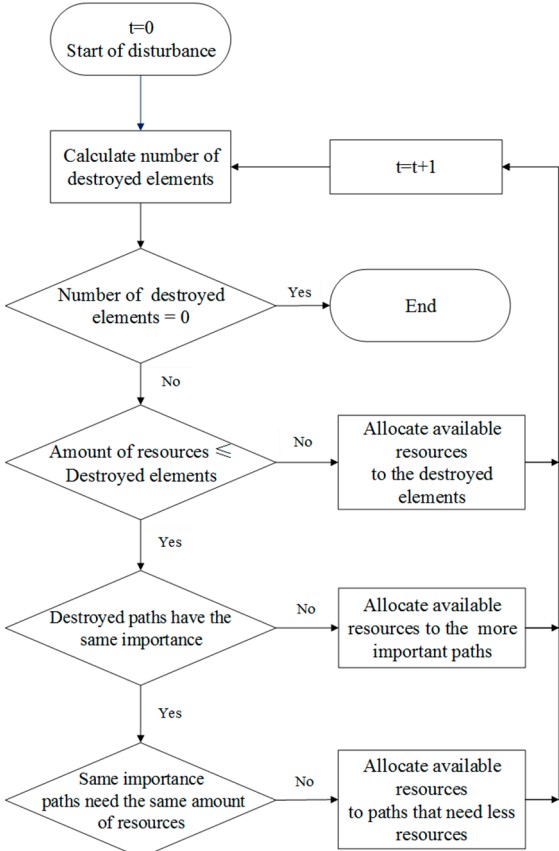

**Figure 11.** Flow diagram of RS-HD.

## 5. Case Study

A case study is developed in this section of the paper in order to evaluate the infrastructure system resilience measure with the five recovery strategies under multiple disturbances.

### 5.1. Multilayer Infrastructure System Network Model

Natural disasters, such as severe weather, earthquakes, hurricanes, floods or antagonistic threats, always strike geographically confined areas. With the geographical location of the infrastructure system components, cell space method is utilized here. To simplify the simulation process, one node of every infrastructure type is assumed to exist in every cell. An example network is shown in Figure 12. The multilayer infrastructure system network contains four layers: Street network, water supply network, power grid and information network. Each layer infrastructure network is modeled, as presented in Section 2.1. Inter-network connections are modeled by attributes of different nodes. For example, the water pump in top-left cell is powered by the electric transmission in the same cell.

The network system includes 16 (4×4) cells. Nodes of single layer networks are numbered from left to right, and from top to bottom. Each node is described by $(\phi, x, y)$, and each edge is defined by two adjacent nodes. Black nodes and edges in Figure 12 are the elements of street infrastructure. Filled black nodes are crossing, and end points connected to outside networks. Black empty nodes are crossing and end points within the network, and edges are street segments. Test network includes $N^S=16$ street crossings and end points, and $E^S=29$ street segments.

Blue nodes and edges in Figure 12 are elements of water supply infrastructure. Blue filled nodes are waterworks (intakes, treatment plants, etc.), blue empty nodes are storage facilities and pump stations, and edges are water transmission pipes. Test network includes $N^w=16$ waterworks, storage facilities and pump stations, and $E^w=17$ water transmission pipes.

Red nodes and edges in Figure 12 are elements of electric infrastructure. Red filled nodes are power plants, empty nodes are elements of the electric transmission system, and edges are electric transmission lines. Test network includes $N^P$=16 power plants and electric transmission nodes, and $E^P$=36 electric transmission lines.

Green nodes and edges in Figure 12 are elements of information infrastructure. Green filled nodes are information control centers, empty nodes are Internet service providers, and edges are cable connections. Test network includes $N^I$=5 Internet service providers, and $E^I$=8 cable connections.

As shown in Figure 12, the filled nodes in water, electric and communication infrastructures provide a source for the system operation. Differently, the black filled nodes have the same function as other black empty nodes. There is an implicit assumption: No function dependence among street segments.

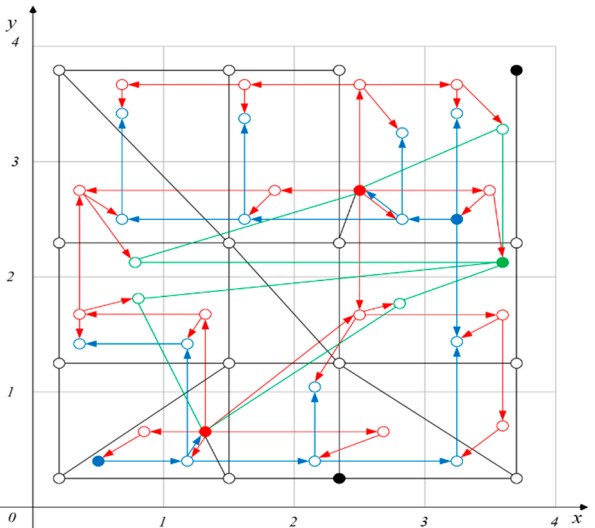

Nodes and edges with blue, red, black and green represent water, electric, street and communication infrastructures respectively.

**Figure 12.** Test infrastructure system network.

*5.2. Infrastructures Interdependence Formulation for the Test Case Study*

This research and the test developed for evaluating a proposed resilience measure include five basic dependence patterns, always present among diverse infrastructure types.

(i)  *Street-water supply network*: Nodes of street and water supply networks located in the same geographic area, which means geographical dependence and use of Equation (5).

(ii)  *Street-power grid*: Nodes of street and power supply networks located in the same geographic area have geographical dependence and use of Equation (5). Power plants need fuel (such as coal) from outside, which means node-path dependence pattern and use of Equation (4).

(iii)  *Water supply-power grid*: Water supply facilities, including waterworks, storage facilities and pump stations are powered by the eclectic infrastructure located in the same geographic area, which means node-node dependence and use of Equation (1). Power plants are provided with water by the water supply facilities in the same geographic area, which means node-edge dependence and use of Equation (2).

(iv)  *Street-information infrastructure network*: Nodes of street and infrastructure networks are located in the same geographic area, which means geographical dependence and use of Equation (5).

(v)  *Power grid-information infrastructure network*: Every electric transmission control depends on the information infrastructure (SCADA systems and similar), which refers to node-node dependence and use of Equation (1). The electric transmission will not work if its links with information infrastructure are cut off, which refers to node-path dependence and use of Equation (4). All the electric transmissions malfunction if the information control infrastructure malfunctions, which refers to the

node-cluster dependence and use of Equation (3). Information control center and Internet service providers are powered by the electric infrastructure located in the same geographic area, which refers to the node-node or node-edge dependencies and use of Equation (1) or (2).

(vi) *Water supply-information infrastructure network*: Nodes of water supply and information networks are located in the same geographic area, which means geographical dependence and use of Equation (5).

In the test example, elements, including nodes and edges in a network, have the same importance, which means that the service level of the same type of infrastructure is the same. The buffering time $T_B$ and the repair time $T_R$ for every infrastructure are assumed to be one day. Though different sectors have different types of resources, it is assumed that one unit of resources have the same effectiveness in repairing one infrastructure system component. For simplification, every sector is assumed to have two units of resources that can be used at the beginning.

## 5.3. Test Case Disturbance—Flood Scenarios

In the network theory, disturbance to the infrastructure system is always represented as the removal of nodes and/or edges from the infrastructure system network. Generally, floods strike geographically confined areas. With the geographical location of the infrastructure system components, components of different infrastructure types/network layers in the same cell will be affected simultaneously. Riverine flooding tends to develop slowly and can last for days and weeks. The water usually spreads over a significant area and inundates infrastructure located in flood plains. For the test case demonstration purposes, three flood scenarios are simulated based on the location of waterworks.

The first flood scenario is a *Small Scale Flood (SSF):* Flood occurs once, only a small area is influenced for a short period of time, and not too many elements of the infrastructure network system, located near the water bodies, are submerged and taken from regular operations. In this scenario, infrastructure system elements located in the bottom four cells, with coordinates $\{0 \leq x \leq 4, 0 \leq y \leq 1\}$, are assumed to be affected.

The second flood scenario is a *Large Scale Flood (LSF):* Flood occurs once and affects large area over a longer time. Many infrastructure system elements located far from the water bodies are affected, due to submergence. In this scenario, infrastructure elements located in the four bottom and four right cells, with coordinates $\{0 \leq x \leq 4, 0 \leq y \leq 1\} \cup \{3 \leq x \leq 4, 1 \leq y \leq 4\}$, are assumed to be affected.

The third flood scenario is a *Sequential Floods (SFS):* Two floods occur sequentially, and influence larger area, starting from the area near the water body. The first flood refers to the removal of infrastructure system elements allocated in four bottom cells, with coordinates $\{0 \leq x \leq 4, 0 \leq y \leq 1\}$. The second flood refers to the removal of infrastructure system elements allocated in the four right cells, with coordinates $\{3 \leq x \leq 4, 1 \leq y \leq 4\}$.

## 5.4. Simulation of Test Network System Performance

The test infrastructure network system (shown in Figure 12) performance is simulated using the dynamic infrastructure system resilience metric presented in Section 3.3 for three flood disturbance scenarios described in Section 5.3, and five restoration strategies presented in Sections 4.1–4.5, respectively. The infrastructure resilience is calculated for the whole multilayer infrastructure network system, as well as for each individual type of infrastructure (network layer).

### 5.4.1. Infrastructure Resilience under Small-Scale Flood (SSF)

Performance units of different infrastructure systems are different. System performance of single layer infrastructure networks with five RSs under SSF are presented in Figure 13. It is worth noting that in this test case, the infrastructure system cannot exceed the initial performance level after the disaster as no infrastructure system improvements can be constructed in a short period of time. Therefore, the rapidity and the end of recovery time is determined by the system performance recovering to normal performance level (pre-disaster performance).

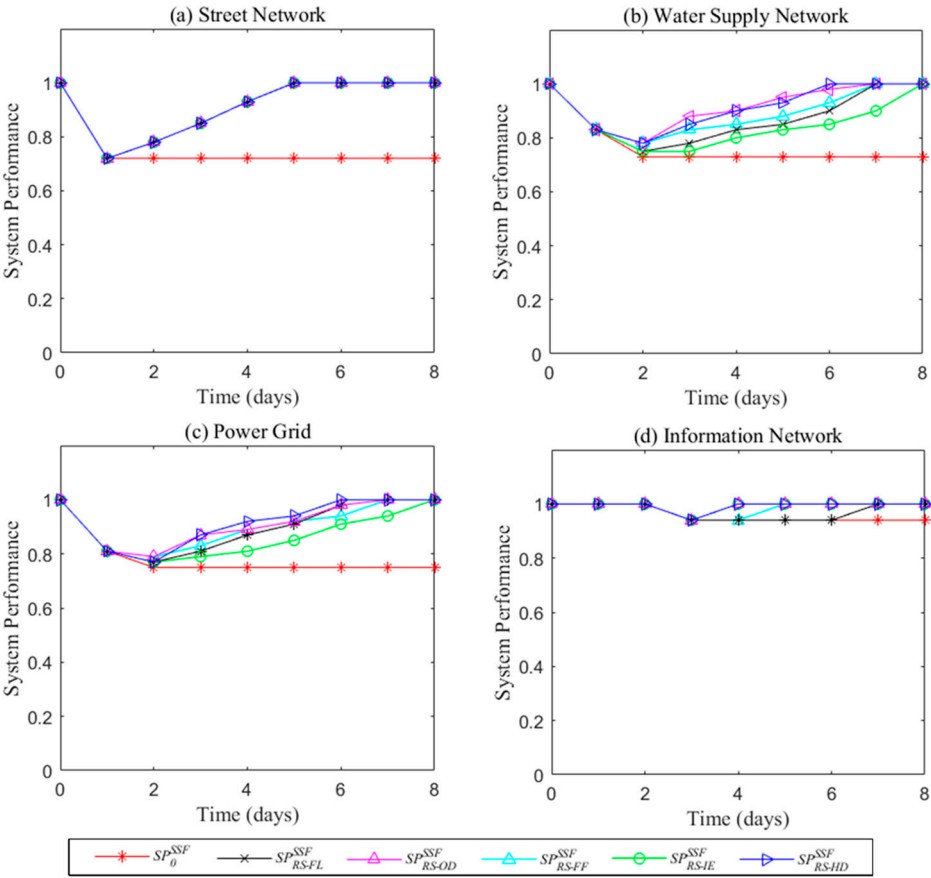

**Figure 13.** Performance with five restoration strategies under the small scale flood scenario.

Though network robustness is a constant under a specific disturbance event, the reactive restorative capacity of infrastructure networks affects the value of rapidity, which differs for different RSs. The impact of five restoration strategies on infrastructure performance can be compared for individual networks. For the street network (see Figure 13a), the system performance curves are the same for all five RSs since there are no functional dependencies between elements within the network; and all elements return to normal operational state immediately after the repair. For water supply network (see Figure 13b), the *RS-OD* and *RS-HD* result in the worst system performance immediately after the disaster. However, in the recovery process, they are outperforming other restoration strategies and lead to faster recovery. For the power grid (see Figure 13c), the performance level achieved with *RS-HD* is always outperforming the other restoration strategies except immediately after the flood disaster. The *RS-OD* results in the second best performance. For the information network (see Figure 13d), there is no direct destruction by the flood event. Some of the elements are affected, due to dependency on the electric energy for maintaining their normal operations. Meanwhile, the *RS-OD* and *RS-HD* are the best choices for quick recovery, and *RS-IE* results in the lowest resilience for all the individual infrastructure systems.

Using Equations (15)–(20), proactive absorptive capacity ($\rho_{PA}^{\phi,SSF}$ and $\rho_{PA}^{SSF}$), reactive restorative capacity ($\rho_{RR}^{\phi,SSF}$ and $\rho_{RR}^{SSF}$), and resilience ($r^{\phi,SSF}$ and $r^{SSF}$) of individual infrastructure networks and multilayer infrastructure network are evaluated. The results are presented in Figure 14. All the infrastructure system resilience under the same disturbance can be compared. In general, proactive absorptive capacities contribute more to system resilience than reactive restorative capacity. Resilience values of the water supply network and power grid are lower than street network and information network, though their proactive absorptive capacities are similar. Multilayer network resilience is determined by the lowest individual single layer network.

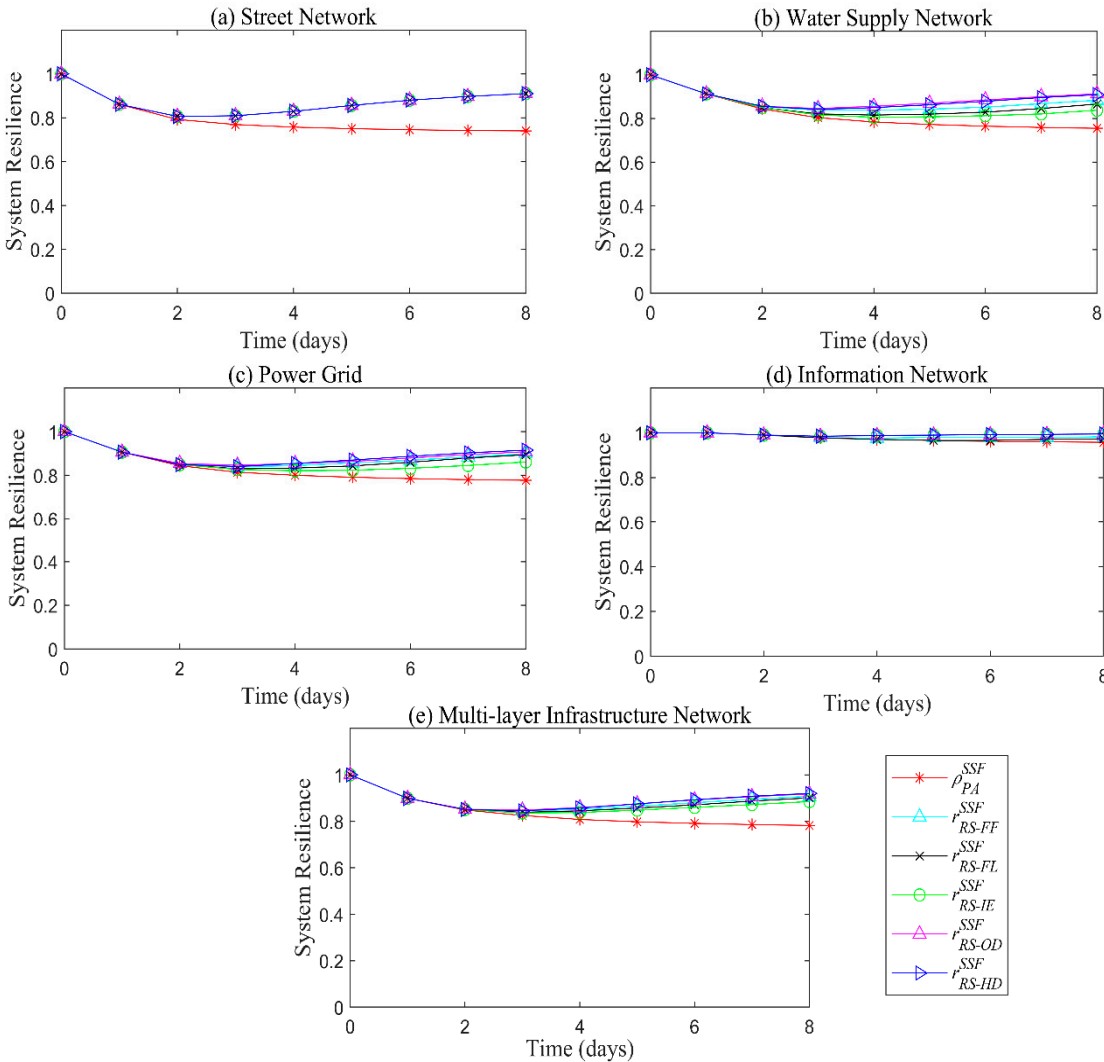

**Figure 14.** Resilience with five restoration strategies under the small scale flood scenario.

Resilience is the integral of the reactive absorptive capacity and the reactive restorative capacity. The bigger value, the more resilient the system. As shown by Equations (17) and (18), the reactive restorative capacity is a function of RS. With the same proactive absorptive capacity under SSF, the different system resilience of the same network is the outcome of different RSs. In the test case, the resilience value for the street network is the same for all RSs, but other types of infrastructure networks show different resilience for different RSs. Generally, infrastructure networks under SSF are showing the highest resilience with the application of *RS-OD*, the fastest recovery with the application of *RS-HD* and the lowest resilience with the application of *RS-IE*. The advantage of these two RSs is that both are taking into account the infrastructure interdependencies.

The detailed analysis of results indicates that the rapidity should be considered for a more comprehensive decision making. For water supply network, the resilience under *RS-OD* is the highest. As the number of destroyed elements is always lower than the number of malfunctioning elements, the rapidity (recovery time) under *RS-OD* is longer than under *RS-HD*. If the average resilience during rapidity is used for comparison, the *RS-HD* strategy results in higher resilience than the *RS-OD,* and the rapidity of the latter is longer. This phenomenon is common for power grid network layer and the entire multilayer network system, which include more interdependencies. Also, the reactive restorative capacity is not only dependent on rapidity, but is also related to the level of resourcefulness.

5.4.2. Infrastructure Resilience under Large-Scale Flood (LSF)

System performance of single layer and multilayer infrastructure networks for five RSs under LFS is shown in Figure 15. Due to the complexity of infrastructure interdependencies, especially in deadlock situation (discussed in Section 6.3), one additional electrical transmission node is added to the test case power grid in order to provide for the network return to a normal operational state.

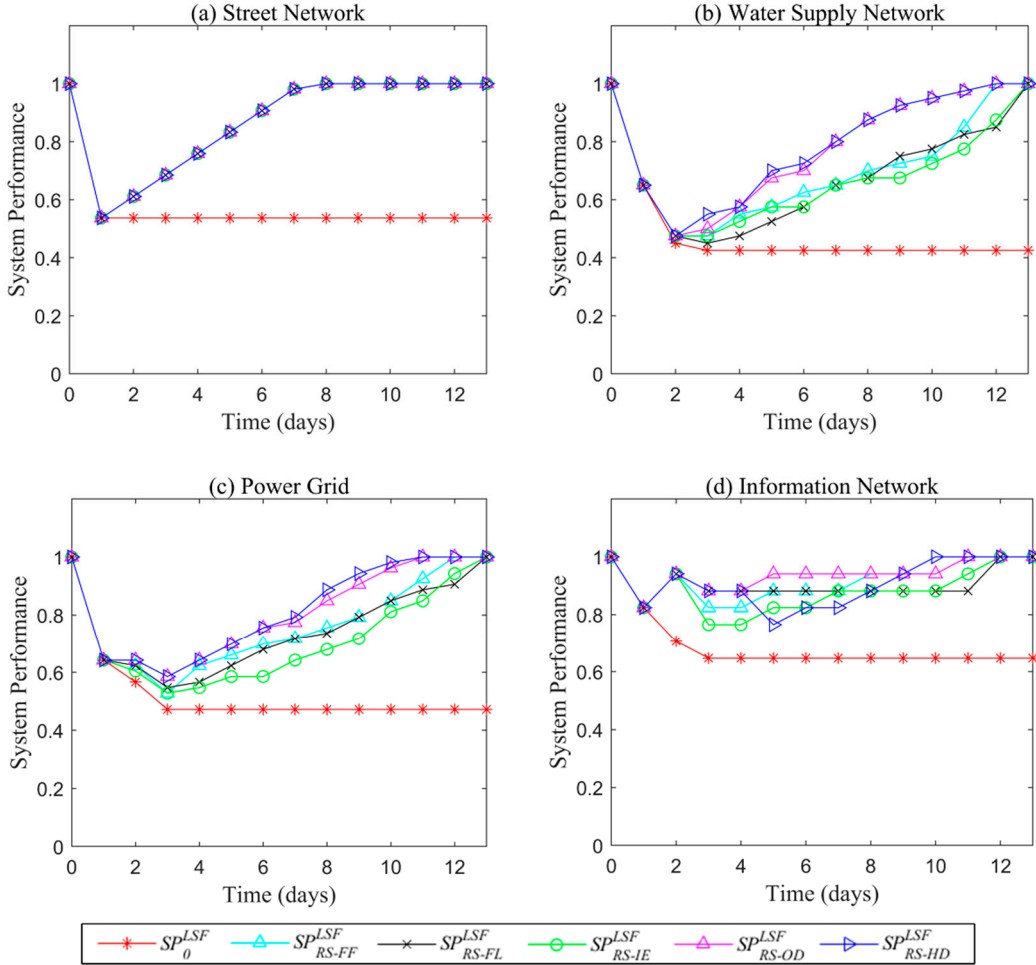

**Figure 15.** Performance with five restoration strategies under the large scale flood scenario.

As shown in Figure 15, under LSF, individual networks and multilayer network system under five RSs perform differently. The street network is an exception (as in the case of SSF). Water supply network under *RS-OD* and *RS-HD* performs obviously better than under other RSs, and *RS-FL* and *RS-IE* result in the worst performance. For the power grid, *RS- OD* and *RS-HD* are the best RSs for rapid restoration of the system to normal operations; the performance of *RS- FF* and *RS- FL* are similar, and *RS- IE* results in the worst performance. The information network, though containing only three elements, is destroyed directly by the LSF and requires twelve days for recovery to normal operation as the interdependencies between information and electric infrastructure networks affected its performance. *RS-HD* strategy leads to the fastest recovery, but not the minimum loss. *RS-OD* provides for one day longer recovery than the *RS-HD*, but always results in better performance.

Figure 16 shows the results of proactive absorptive capacity ($\rho_{PA}^{\phi,LSF}$ and $\rho_{PA}^{LSF}$), proactive restorative capacity ($\rho_{PR}^{\phi,LSF}$ and $\rho_{PR}^{LSF}$), and resilience ($r_{\phi}^{LSF}$ and $r^{LSF}$) of individual infrastructure and multilayer infrastructure networks under large flood scenario.

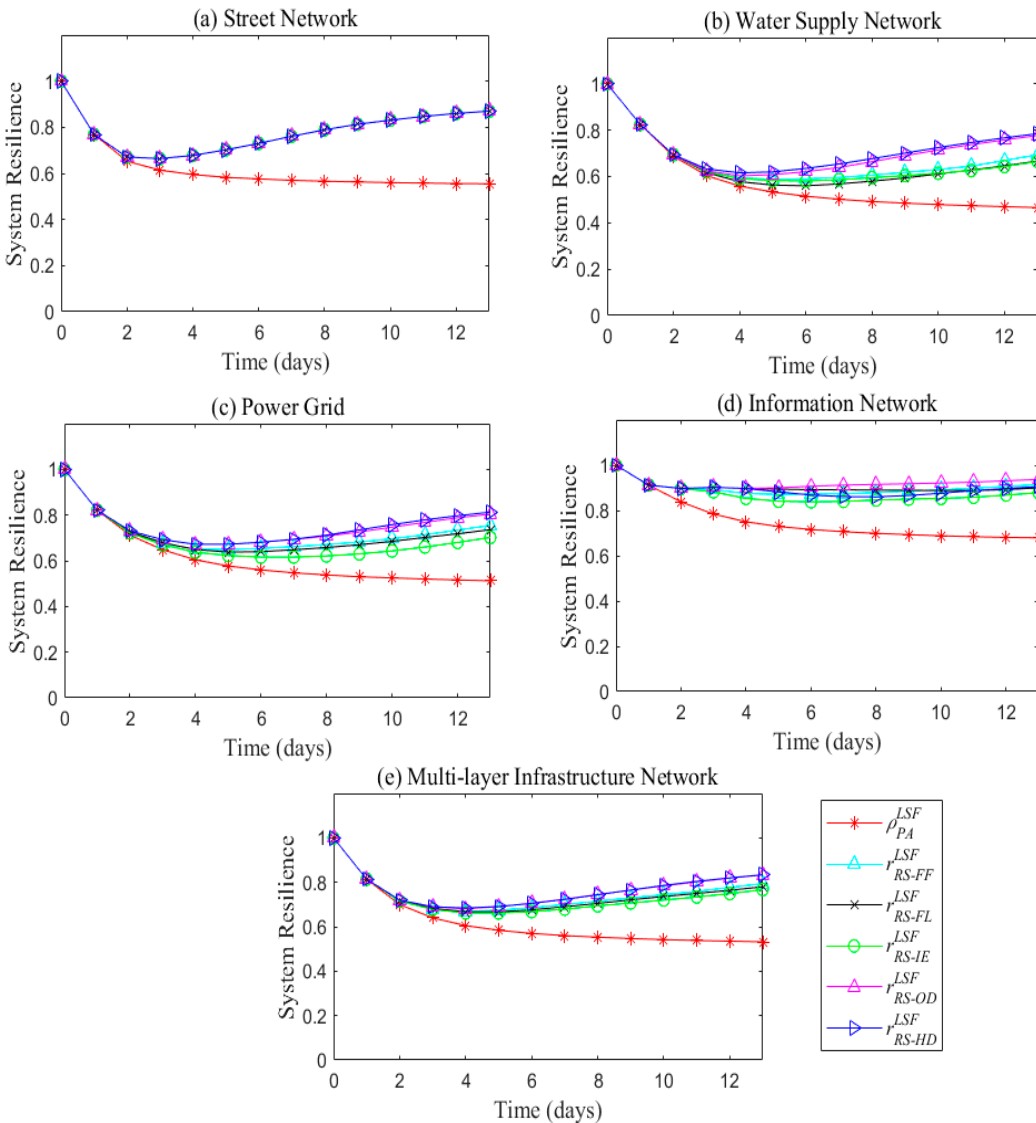

**Figure 16.** Resilience with five restoration strategies under the large-scale flood scenario.

As shown in Figure 16, system resilience for five RSs under the LSF provides more insight into test system performance. Generally, infrastructure systems with *RS-OD* strategy, that are taking into account the infrastructure interdependencies, are showing higher resilience than other RSs. Different from SSF scenario, *RS-HD* shows the highest system resilience for each single layer infrastructure network, as well as for the multilayer network system. For information network, the resilience value with *RS-OD* is higher than with *RS-HD*, but the rapidity with *RS-HD* is shorter. Meanwhile, all the infrastructure networks with *RS-IE* are showing the lowest resilience value.

### 5.4.3. Infrastructure Resilience under Sequential Floods (SFS)

System performance of single layer and multilayer Infrastructure networks with five RSs under SFS are presented in Figure 17. As the first flood event occurs at t=0, and the second at t=2, the performance line for every network layer has two obvious declines. In this scenario, one additional electrical transmission is also added to the power grid to allow the infrastructure system to return to normal operation.

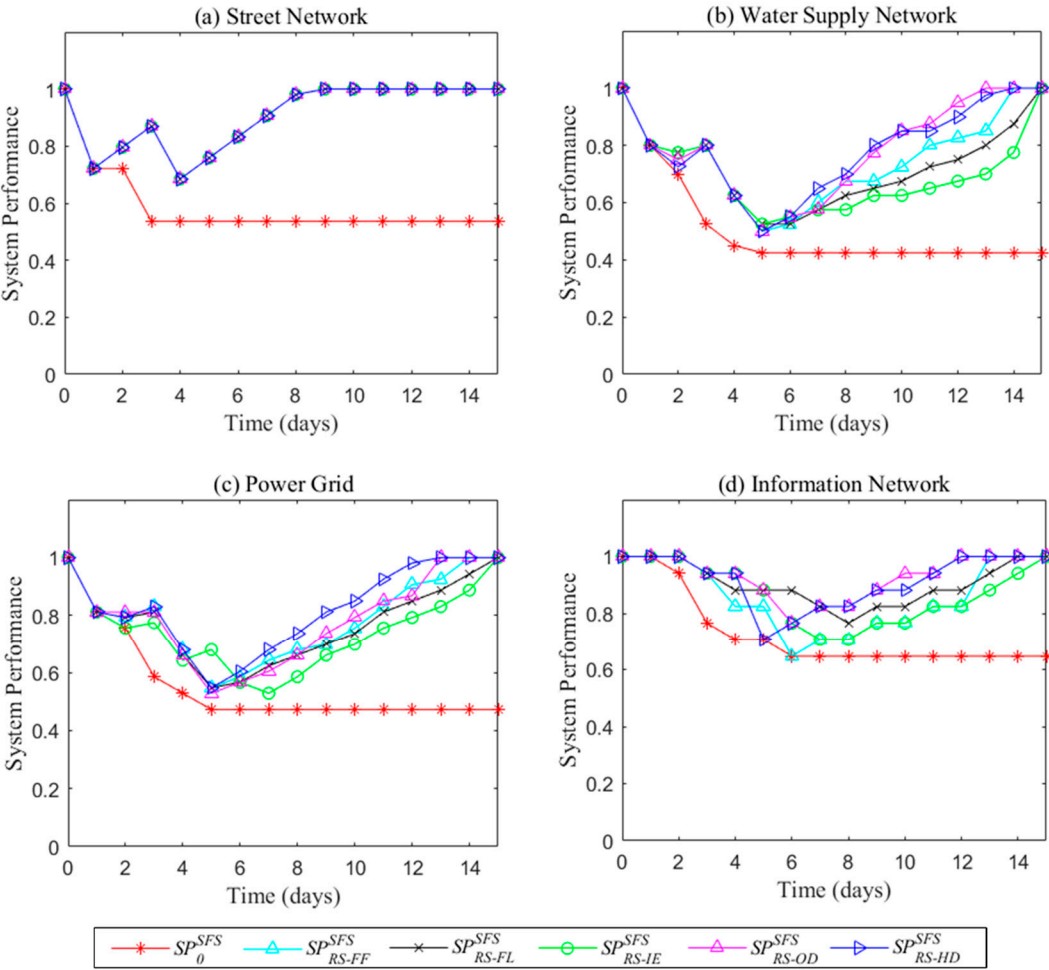

**Figure 17.** Performance with five restoration strategies under sequential flood scenario.

According to Figure 17, networks under RSs that include infrastructure interdependencies (*RS-OD* and *RS-HD*) always exhibits a better performance. For water supply network, though *RS-FL* results in the best performance after the first flood event, *RS-OD* and *RS-HD* leads to faster recovery and lower loss after the second flood event. For the power grid, *RS-OD* results in the best performance after the first flood event, and *RS-HD* leads to the best performance after the second flood event. For the information network, system performance is the same for the five RSs after the first flood event since there is no directed destruction of the infrastructure elements. The *RS-OD* strategy results again in the best performance after the second flood event. Integral performance of multilayer network with *RS-HD* is the worst after the first flood event, but the best after the second. Since there are no interdependencies between the street network and other networks except geographic one, the performance of the street network with all five RSs is the same. Also, all networks under *RS-IE* always show worse performance, especially after the second flood event. Especially the power grid performance under *RS-IE* increases, followed by a decrease at t=5 days, when other single layer networks reach the minimum performance levels. This is a typical disadvantage of the *RS-IE*, which illustrates that the electric transmission will continue malfunctioning after repair, due to inter-network dependencies.

For the results illustrated in Figures 13, 15 and 17, the robustness of network system—the number of elements without loss of function (not affected by the disturbance)—is calculated using Equations (8) and (9). Robustness is a system's inherent property, relating infrastructure system structure and scale of the disturbance, and could not change within a short period of time. In spite that the duration of disturbance event is assumed to be one day, the impacts of disturbance will cause loss of system

performance over a longer period of time, due to dependencies among system elements. Therefore, the robustness level of most networks is reached at the latter time.

Figure 18 shows the results of the reactive absorptive capacity ($\rho_{PA}^{\phi,SFS}$ and $\rho_{PA}^{SFS}$), proactive restorative capacity ($\rho_{RR}^{\phi,SFS}$ and $\rho_{RR}^{SFS}$), and resilience ($r_{\phi}^{SFS}$ and $r^{SFS}$) of individual infrastructure and multilayer infrastructure networks under a sequential flood scenario.

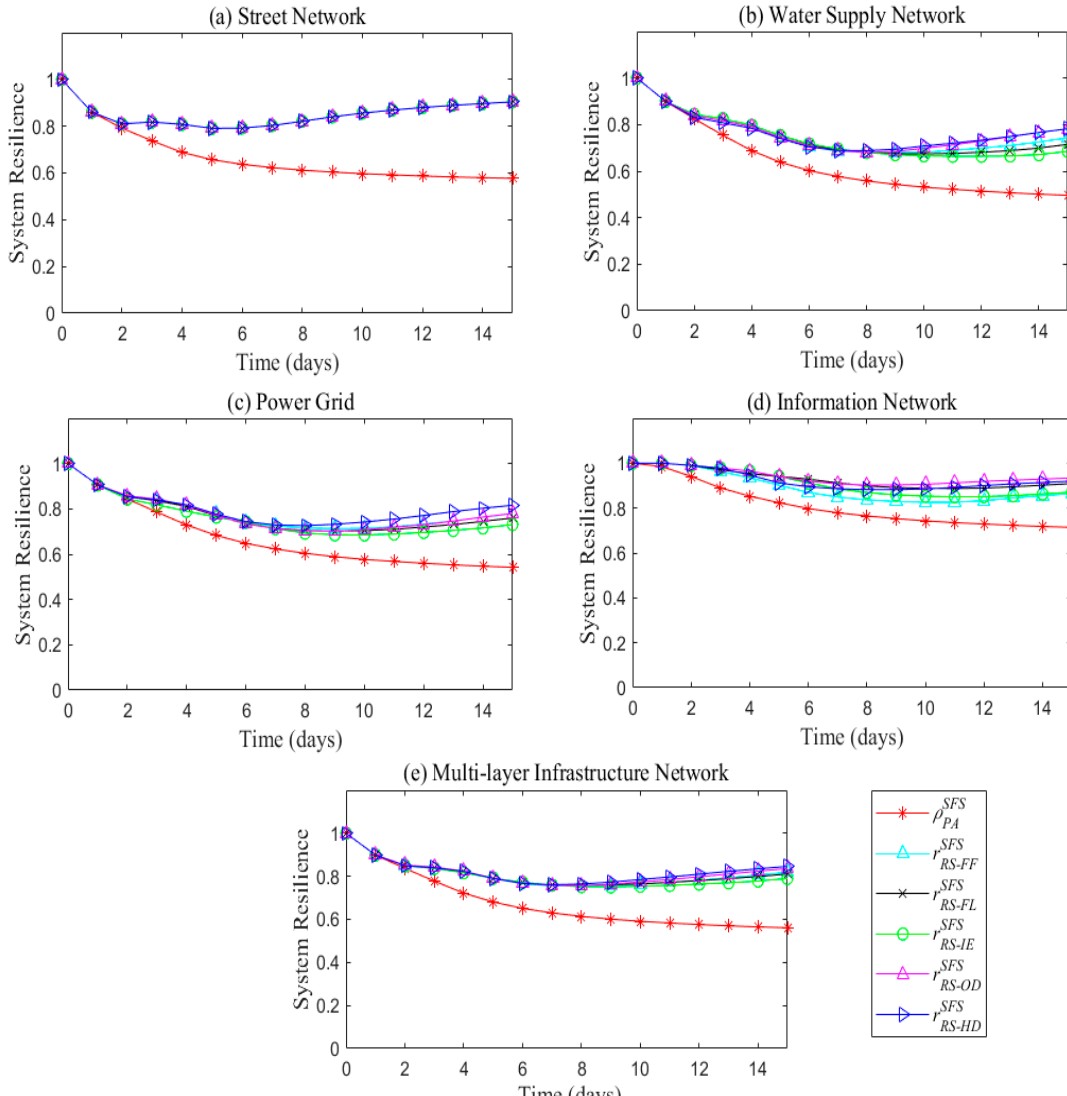

**Figure 18.** Resilience with five restoration strategies under sequential flood scenario (SFS).

According to Figure 18, relative merits of the five RSs according to system resilience are not always consistent. Especially after the second flood event, single layer and multilayer networks resilience with the five RSs are slightly different. After five days of the second disturbance event, all infrastructure networks, except information network, under *RS-HD* are more resilient than under any other restorative strategy. Therefore, the infrastructure system with *RS-HD* is the most resilient, with *RS-OD* is the most rapid, and with *RS-IE* is the least resilient.

Comparison of Figures 16 and 18 shows different resilience of infrastructure networks in spite of the fact that the sum of direct physical losses for the SFS scenario is the same as for the LSF scenario. Duration of disturbance has an influence on the infrastructure recovery. The average rapidity under SFS scenario is 15 days, which is longer than that of 12 days under the LSF scenario. Moreover, *RS-HD* results in a more resilient system than *RS-OD* and other RSs.

## 6. Discussion

The direct physical loss by the flood scenarios is always lower than actual functional losses, as shown in Figures 13, 15 and 17, and is independent of the scale or duration of the disturbance event. The main reason for that is the existence of interdependencies between the three infrastructure (water, energy, and information) impacts, resulting in more severe overall consequences than those from direct impacts of the primary disturbance event.

### 6.1. Hidden Impacts

Based on the system performance and resilience of single layer and multilayer infrastructure networks illustrated in Figures 13–18, *RS-HD* always leads to better results. *RS-HD* refers to the restoration strategy giving repair priority to elements of hidden dependent patterns or path dependent patterns, which are represented by Equation (3). This kind of infrastructure dependence is invisible in normal operations, but emerges and becomes obvious under disturbance. It can also easily result in huge losses, so-called hidden impacts. One example could be the path dependence between the power plant and its connections with outside through the street network. Therefore, under the LSF, the performance of power grid decreases sharply as no power generation source (coal, oil, gas) is not available due to the affected connections with the outside world. Therefore, the first repair of all the elements of this dependent path would lead to faster recovery of the infrastructure system and increase system reactive restorative capacity. That is the reason why the *RS-HD* strategy always leads to more resilient infrastructure system.

Generally, hidden dependencies do not exist only among different types of infrastructure, but may also be present within individual network types. They also called functional dependencies. For example, the normal operation of an electric transmission node is dependent on its connections with a power plant and an information control system. Normal operation of a water pump is dependent on its connection with waterworks and a power plant, etc. Thus, the hidden impacts here are best illustrated with physical functional dependencies and cyber dependencies. *RS-HD*, in this paper, takes both intra-network and inter-network hidden dependencies into account, and gives priority to the inter-network elements.

### 6.2. Chain Impacts

Chain impacts refer to the failure of diverse infrastructure elements caused by the chain dependence among them, which combines node-node, node-edge, or node-cluster infrastructure dependences, Equations (1), (2), and (4), respectively. According to system dynamics mechanism and simulation results, chain impacts influence infrastructure resilience in three ways.

(i) Loss expansion—malfunction of information infrastructure elements causes malfunction of controlled electric infrastructure, and leads to malfunction of powered water infrastructures (as shown in Figure 19).

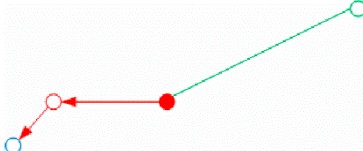

**Figure 19.** Chain impacts of diverse Infrastructure system (part of Figure 12).

(ii) Loss continuity—the disturbance impacts are extended by a period of time after the disturbance end time. The maximum loss of system performance for a single layer or a multilayer infrastructure network is always delayed for a period of time. For example, as the LSF happens at t=0, the lowest level of system performance of the water supply network is reached at t=3 with *RS-FL* or t=2 with other RSs. The similar results are visible in Figures 13, 15 and 17.

(iii) Integrated multi chain impacts—multi chain impacts integrated within network dependencies. For example, under SSF the information infrastructure malfunction results in malfunction of controlled electric transmission elements and then powered water infrastructure elements is also affected. In the same time, the malfunctioned electric transmission elements cause power information infrastructure malfunction because the electric transmission is cut off. The combined two chain impacts lead to information infrastructure malfunction for a long period of time, as it can be seen in Figures 13, 15 and 17.

So, the infrastructure systems with RSs that include infrastructure interdependencies are always more resilient, as it can be seen in Figures 14, 16 and 18.

### 6.3. Cycle impacts

The cycle impact is described by disturbance of one infrastructure type that propagates to other dependent infrastructure and back to the infrastructure where the failure originated. Cycle impacts always cause the system dead lock. This behavior is a difficult and common problem in practice, and is the reason for adding to the test case system an electric transmission component for system recovery under LSF and SFS. The cycle impacts among electric plant, electric transmission, waterworks and water pipes lead to infinite malfunction loop under LSF and SFS. An illustration of this effect is in Figure 20.

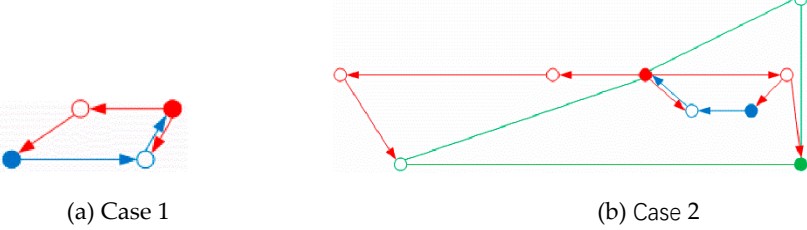

(a) Case 1　　　　　　　　　　　　　　　　　(b) Case 2

**Figure 20.** Cycle impacts a diverse infrastructure system (parts of Figure 12).

As shown in Figure 20a, blue nodes and edges are water supply infrastructure elements, where the filled blue node is a waterworks, the empty blue node is a pump station, and blue lines are water transmission pipes. Filled red node is a power plant, the empty red node is an electric transmission element, and red lines are electric transmission lines. Water infrastructure needs a steady supply of electric energy to maintain its normal operations, while the electric power plant requires the provision of water for normal power generation (for example, for cooling). If all the infrastructure elements illustrated in Figure 20a are directly destroyed by LSF or SFS, and then the nine infrastructure elements (three water infrastructure elements and five electric infrastructure elements in the test case network) are repaired, all of them will continue malfunctioning as no electric power is available to activate the waterworks or no water is available to activate the power plant. To solve this problem, an additional electric resource is needed until all the destroyed infrastructure elements within the cycle are repaired.

The cycle impacts not only exist between water and electric infrastructure elements, but also between water, electric and the information infrastructure elements. They make the restoration process much more complex and longer. Figure 20b illustrates the typical cycle impact among the three types of infrastructures under LSF and SFS. Therefore, the repair of destroyed elements as soon as possible not only leads to faster recovery of the whole system, but also prevents cascading failure and increasing loss of performance. That is why the same direct loss scale of LSF and SFS with same RS results in different system resilience. Also, the combination of three impacts: Hidden impacts, chain impacts and cycle impacts, is the reason why the *RS- IE* strategy is always the worst for any infrastructure networks.

## 7. Conclusions

Disaster is a reality and is threatening all three pillars (social, environmental, economic) of sustainable development. Human activity grows exposure, increasing the propensity for systems reverberations, setting up feedback loops with cascading consequences that are difficult to foresee. Resilient infrastructure system is essential for the continued operation of a city, a region and a country. While more and more organizations, domestic and international, public and private are promoting infrastructure resilience to lower disaster tolls, transforming this important concept to practice remains a work in progress. Domestic specific emergency response plans of individual infrastructures are not well integrated with the others. To reduce the multi-hazards risk and ensure the whole system sustainability, this research tries to promote an interdependent infrastructure system resilience enhance methods.

Resilience measure is presented as a quantitative approach for complex infrastructure system management that provides for effective evaluation and improvement of normal and emergency infrastructure system operation capacities. This paper is using the network theory for the development of an original infrastructure resilience model that takes into account infrastructure interdependencies. This model is not only a framework for system-of-systems resilience analysis, but also provides an evaluation method of various protection and restoration strategies that will optimize the performance of interdependent infrastructure system.

With a numerical test, the simulation results show that the sector-specific decisions could not always lead to optimal system solutions, i.e., infrastructure systems with *RS-IE* always end with the lowest resilience. In the paper, the amount and effectiveness of resources are assumed the same among different sectors. This limits the results of the analyses and points out that restoration resources should not be evenly distributed, but based on the sector's contribution to system-level resilience improvement. Therefore, the resilience model can also be used for planning and distribution of restoration resources.

This paper simplifies the real infrastructure system structure and uses a numerical test for illustrative purposes. However, it is very clear that the framework can be easily extended to evaluate system resilience under other types of disasters. Social, economic and organizational resilience can also be analyzed with the community and organization management data provided. Thus, the framework can be used by the municipal decision makers for a community resilience improvement under various types of disasters by development and timely preparation of disaster management emergency plans.

Finally, actual infrastructure systems are of high complexity. Proactive absorptive capacity and reactive restorative capacity interact through changing robustness and resourcefulness of the infrastructure system. The numerical test presented in the paper only takes the latter into account. In addition, this paper only considers cascading failures caused by the near physical failure of elements. Addressing the malfunctions generated by the flow overload along the path and analyzing their impacts on system restoration processes and resilience are topics for further research. The future research will also focus on the interaction of the reactive absorptive capacity and proactive restorative capacity in a real case study.

**Author Contributions:** Conceptualization, J.K.; methodology, J.K.; software, C.Z.; validation, J.K.; formal analysis, J.K.; investigation, S.P.S.; resources, J.K.; data curation, C.Z.; writing—original draft preparation, J.K.; writing—review and editing, S.P.S.; visualization, C.Z.; supervision, S.P.S.; project administration, J.K.; funding acquisition, S.P.S.

**Funding:** This research was funded by the Natural Sciences and Engineering Research Council (NSERC) of Canada, the Natural Science Foundation of China under Grant No. 71704111, Shanghai Science and Technology Development Funds under Grant No. 19692107300, 19ZR1417300, and 18ZR1413200, National Key Research and Development Project of China under Grant No. 2018YFB1503100, Science and Technology Project of State Grid Corporation of China under Grant No. SGJSJY00GHJS1900066, and Institute of Free Trade Zone of SUFE and Shanghai Education Committee under Grant No. 2019110183.

**Conflicts of Interest:** The authors declare no conflict of interest.

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
