# Peer review of "Resilience Assessment of Interdependent Infrastructure Systems: A Case Study Based on Different Response Strategies"

_sustainability, doi:10.3390/su11236552_

Round 1
Reviewer 1 Report
Resilient infrastructure systems is an interesting and actual theme. The authors approached, the topics were presented appropriately. All necessary items were described. Also, the article was written clearly and comprehensively. I suggest that the authors improve the quality of figures 13, 14, 15, 16, 17 and 18.
Reviewer 2 Report
The paper titled: “Resilience Assessment of Interdependent Infrastructure Systems: A Case Study Based on three Different Response Strategies” contributes in a new infrastructure network resilience measure capable of addressing complex infrastructure system, as well as network component interdependency.
The assumption has been supported by means of a quantitative resilience measure using dynamic space-time simulation model. The topic is interesting and the structure of the paper appropriate, in my opinion the paper is suitable for publication.
Please describe with some example the five restoration strategies applied in the study. A more extensive captions must be added to help readers to clearly understand the content of the figures (eg. 14-15-16); Please specify why for some layer of infrastructure only few restoration strategies are shown. Can please the author explain the added value to this approach? Who can be the final user and explain which the benefit in the real application are?Author Response
Please see the attachment.

Reviewer 3 Report
It is a great job but not suitable for the journal.
It is very specific work on structural analysis, forgetting the concept of sustainability (that is composed of aspects of economics, environmental and social).
I suggest submitting the paper to another journal.
Round 2
Reviewer 3 Report
The authors have made the requested changes
Author Response
Point 1 :The authors have made the requested changes.
Response: Thank you for your comment.